

# Validation of SMAP L2 passive-only soil moisture products using *in situ* measurements collected in Twente, The Netherlands

Rogier van der Velde[1], Andreas Colliander[2], Michiel Pezij[3], Harm-Jan F. Benninga[1], Rajat Bindlish[4], Steven K. Chan[2], Thomas J. Jackson[5], Dimmie M.D. Hendriks[6], Denie C.M. Augustijn[3], and Zhongbo Su[1]

[1] Department of Water Resources, University of Twente, Enschede, 7500 AE, The Netherlands
[2] Jet Propulsion Laboratory, California Institute of Technology, Pasadena, CA 91109, USA
[3] Department of Water Engineering and Management, University of Twente, Enschede, 7500 AE, The Netherlands
[4] NASA Goddard Space Flight Center, Greenbelt, MD 20771, USA
[5] Hydrology and Remote Sensing Laboratory, USDA ARS, Beltsville, MD 20705, USA (Retired)
[6] Department of Subsurface and Groundwater Systems, Deltares, Utrecht, 3508 AL, The Netherlands

*Correspondence to*: Rogier van der Velde (r.vandervelde@utwente.nl)

**Abstract.**

The Twente region in the east of the Netherlands has a network with twenty soil monitoring stations that has been utilized for validation of the Soil Moisture Active/Passive (SMAP) passive-only soil moisture products. Over the period from April 2015 until December 2018, seven stations covered by the SMAP reference pixels have fairly complete data records. Spatially distributed soil moisture simulations with the Dutch national hydrological model have been utilized for the development of upscaling functions to translate the spatial mean of point measurements to the domain of the SMAP reference pixels. The native and upscaled spatial soil moisture means have been adopted as *in situ* references to assess the performance of the SMAP i) Single Channel Algorithm at Horizontal Polarization (SCA-H), ii) Single Channel Algorithm at Vertical Polarization (SCA-V), and iii) Dual Channel Algorithm (DCA) soil moisture estimates. In the case of the Twente network it was found that the SCA-V soil moisture retrieved SMAP observations collected in the afternoon had the best agreement with the *in situ* references leading to an unbiased Root Mean Squared Error (uRMSE) of 0.059 $m^3$ $m^{-3}$. This is larger than the mission target accuracy of 0.04 $m^3$ $m^{-3}$, which can be attributed to large over- and underestimation errors (>0.08 $m^3$ $m^{-3}$) in particular at the end of dry spells and during freezing, respectively. The strong vertical dielectric gradients associated with rapid soil freezing and wetting causes the disparity in soil depth characterized by SMAP and *in situ* that leads to the large mismatches. Once filtered for frozen conditions and antecedent rainfall the uRMSE improves to 0.043 $m^3$ $m^{-3}$.

## 1 Introduction

Science and professional communities rate the success of satellite missions, at least in part, by error metrics calculated between its data products and a reference. ESA's Soil Moisture and Ocean Salinity (SMOS, Kerr et al. 2001) and NASA's Soil Moisture Active Passive (SMAP, Entekhabi et al. 2010) mission apply a target accuracy of 0.04 $m^3$ $m^{-3}$ to the retrieval of soil moisture from L-band brightness temperature in the top 5 cm of soil. Comprehensive calibration and validation programmes (e.g. Jackson et al. 2013, Delwart et al. 2010) are, therefore, imperative to establishing a reference to which the retrieved soil moisture can be compared in an objective manner. This is, however, complicated by the disparity in spatial support provided by satellite observations and the techniques available to measure soil moisture *in situ* (Western and Blöschl 1999, Cosh et al. 2015). Indeed, the area covered by measurements taken with most ground sampling techniques is a mere fraction of the area covered by satellite footprints (Jackson et al. 2012). Also, differences between soil depths that can be sampled reliably *in situ*



and observed with microwave radiometry contribute to this spatial-scale mismatch (e.g. Escorihuela et al. 2010, Zheng et al. 2019).

A strategy that can be adopted for validation of satellite-based soil moisture estimates is accepting the spatial-scale mismatch and focusing efforts on quantifying spatial sampling errors by introducing a third independent data set of the same quantity (e.g. Stoffelen et al. 1998, Miralles et al. 2010); referred to as Triple Collocation (TC). Recently, Chen et al. (2017) applied its classic and extended TC versions for validating SMAP L2 data products using data from sparse *in situ* monitoring networks and acknowledged its merits in correlation-based assessments, but they found that TC is unlikely able to correct RSMEs (root mean squared errors) for spatial sampling errors due to large biases between point and satellite-based estimates. This supports the choice for using sites with multiple measurement locations inside satellite footprints as the primary basis for assessing the performance of the SMAP L2 data products (Chan et al. 2016, Colliander et al. 2017).

However, even with a network with a multitude of soil moisture monitoring stations, the question remains how to translate the collection of spatially distributed point measurements to the satellite footprint and how many stations are needed to establish a reliable spatial mean. Crow et al. (2012) provide a review of the upscaling of point measurements to satellite footprints. For instance, Famiglietti et al. (2008) applied stochastic analyses to surface soil moisture data sets collected during intensive field campaigns, and concluded that 30 samples are needed to estimate the spatial mean of 50 x 50 km$^2$ domains (approximate size of satellite footprints) with an accuracy of 0.03 m$^3$ m$^{-3}$ with 95 % confidence. Brocca et al. (2010) used statistical and temporal stability analyses applied to year-round intensive field measurements to conclude that nearly 40 samples are needed to determine the catchment (60 x 60 km$^2$) mean soil moisture with an accuracy of 0.02 m$^3$ m$^{-3}$. Other investigators (Mohanty and Skaggs 2001, Cosh et al. 2005, Martínez-Fernández and Ceballos 2005) have adopted the temporal stability concept to diagnose how well measurements taken at specific locations represent the spatial mean recorded by the network. However, De Lannoy et al. (2007) indicated that temporal stability may be less suited as upscaling strategy under complex hydrologic terrain with a strong soil layer stratification and/or spatially heterogeneous water management practices. Moreover, reliance on a limited number of measurement sites makes the spatial mean vulnerable to spatially varying rainfall and the continuity of the data streams.

In summary, preferably more than 30 spatially independent measurements are suggested to establish a reliable reference for validation of satellite-based soil moisture estimates and no generally accepted upscaling protocol exists with which metrics can be derived without spatial sampling errors. This poses a challenge on validation activities as only a few monitoring networks with such high density of stations are available at a particular time mainly because of the difficulty in keeping all stations operational. This led to accepting an accuracy of 0.03 m$^3$ m$^{-3}$ for the spatial mean with a 70 % confidence, implying a requirement of at least eight monitoring stations within satellite footprints, and the application of an upscaling strategy based on weights assigned to individual measurements using Thiessen polygons (Colliander et al. 2017).

The Twente soil moisture monitoring network, located in the central-eastern part of the Netherlands, is one of the regions that has been used for validation of the SMAP L2 passive-only soil moisture products (Colliander et al. 2017, Chan et al. 2018). The Twente network consisted of twenty locations where both soil moisture and temperature were measured at various depths.



However, not all stations could be utilized because some did not fall within the satellite validation footprint and other stations did not consistently deliver data as result of instrument failures, vandalism or changes in landownership. As a result, the requirement of eight operational stations was not always met. This was in particular the case for April to October 2015. Hence, a model-based upscaling strategy, similar to the one described in Crow et al (2005), was explored to be able to utilize the in-situ measurements for validation of the SMAP L2 soil moisture estimates also for periods when the number of operational stations would fall short of the requirement.

In this paper, we report on the development of a model-based upscaling method with scaling parameters derived directly from the mean and standard deviation of the *in situ* measured and model simulated soil moisture. We adopted the Dutch integrated water resources model (De Lange et al. 2014), called 'Landelijke Hydrologisch Model' (LHM, National Hydrological Model in Dutch) that simulates the transfer of water masses across the groundwater, unsaturated zone and surface water reservoirs. LHM simulated soil moisture matching *in situ* measurements from January 2015 till October 2018 were utilized to develop the upscaling functions, which were subsequently used to assess the performance of the SMAP L2 passive-only soil moisture product for the periods from April 2015 to December 2018.

## 2. Study area and *in situ* measurements

### 2.1. Twente region

The monitoring stations are distributed across the central-eastern part of the Netherlands in the region known as 'Twente'. Characteristics of this region include very little relief and the use of land for farming practices amongst various mid-sized urban conglomerates. Pastures make up for more than seventy percent of the agricultural fields and the remainder is mostly corn, wheat and potatoes. The growing season generally starts in April and ends around October/November depending on the weather and crop type. In this period, the pastures are either grazed by cattle or mowed four to six times. Forested areas are typically found on the gently rolling hills (< 50 m in height).

Sandy, loamy, and peat soils are the main soil types (Dente et al. 2011). Sand and loamy sand are dominant near the surface and some peat remnants are spread across the domain. The subsurface consists mostly of glacial windblown sand deposits in the south and poorly drained soils in the north. The groundwater tables are rather shallow across the entire region; close to the surface during winters and 1 to 2 meters deep during summers. Water standing on the fields can be persistent in low-lying areas in the cold wet season and may also occur in the warm season after intensive rain.

Climate in Twente is determined by a mixture of influences from the sea towards the west and land towards the east. Moist and warm (or cold in the summer) air is received from the sea direction, whereas dry and cold (or warm in the summer) air is blown from land. As a result, both winters and summers are generally mild and wet. However, fairly extreme cold and warm episodes may occur depending on the wind direction with temperatures well below -10 ℃ in winter or well above 30 ℃ uin summer, respectively. Figure 1 provides an impression of the prevailing weather conditions for the study period, from January 2015 till February 2019, by showing monthly averages of daily mean, minimum and maximum 1.5 m air temperatures ($T_{\text{mean}}$,





$T_{min}$ and $T_{max}$, respectively) along with monthly rainfall and reference crop evapotranspiration ($ET_{ref}$) sums. Data presented in the plot are derived from meteorological measurements collected at the three weather stations in the region (see Fig. 2) operated by the Royal Netherlands Meteorological Institute (KNMI). It should be noted that 2018 was an exceptional year with record high temperatures and evaporative demand, and low rainfall.

## 2.2. Measurement networks

Figure 2 displays the distribution of soil monitoring stations and KNMI weather stations and the extent of the SMAP reference pixels laid over a Landsat 8 true colour composite of 20 July 2013. Wind, temperature, radiation, precipitation, visibility, humidity and $ET_{ref}$ observations for the Heino and Hupsel automated weather stations are available from 1989 onwards, and the Twenthe record even dates back to 1951. More information and data access can be obtained from http://www.knmi.nl/nederland-nu/klimatologie (last access: July 10th, 2019).

The University of Twente (UT-ITC) soil monitoring network was developed in 2009. Twenty stations were setup to measure soil moisture and soil temperature every fifteen minutes with Decagon EC-TM ECH2O probes installed at nominal depths of 5, 10, 20, and 40 cm below the surface. Budget constraints restricted the number of fully equipped stations, but every station included a sensor at 5 and 10 cm. A soil-specific calibration function was developed for the EC-TM probe under laboratory conditions with which an estimated accuracy of 0.023 $m^3$ $m^{-3}$. Readers are referred to Dente et al. (2012) and Dente et al. (2011) for additional information on the network development.

Over time, the original location of a number of stations had to be changed for various reasons (e.g. vandalism, land use/ownership change). This and equipment failures caused gaps in the data records. In the fall of 2015, a concerted maintenance operation was undertaken to update the network. Activities included reinstallation of stations within the same field, in an adjacent field of the same landowner or in a neighboring parcel of a different landowner. The newly installed stations are equipped with five new Decagon 5TM probes at nominal depths of 5, 10, 20, 40 and 80 cm with the exception of station 04 (three probes) due to a shallow groundwater table and station 07 (four probes). Also, a new soil-specific calibration function was developed for the 5TM probe following the same procedure as described in Dente et al. (2011) with an accuracy of 0.027 $m^3$ $m^{-3}$ expected. Since the fall of 2017, stations 06, 19 and 20 are no longer operational and have been replaced by equipment installed at the premises of the KNMI weather stations Hupsel and Twenthe. Table S1 lists the stations, the vegetation cover, the sensor types and depth, and dates on which the locations of the stations were changed, when applicable.

## 3. Spatial data

### 3.1. SMAP and its passive-only soil moisture products

The SMAP mission includes a radiometer that measures at a centre frequency of 1.4135 GHz (L-band) the four Stokes parameters of which two provide the horizontally (H) and vertically (V) polarized brightness temperatures ($T_b$), and the other two ($T_3$ and $T_4$) channels serve to correct for the Faraday rotation. Additional signal processing techniques have been embedded



within the design of SMAP to assist with the detection and mitigation of RFI (see Piepmeier et al. (2014) and Mohammed et al. (2016) for details). The SMAP radiometer observes at an angle of 40° with a 3-dB instantaneous field of view (IFOV) of 38 km x 49 km on the earth surface from a sun-synchronous orbit every 2 to 3 days at nominal overpass times of 6:00 AM (descending) and 6:00 PM (ascending) near the equator (Piepmeier et al. 2017). All native gridded radiometer-based data products are placed on an Equal-Area Scalable Earth (EASE) grid projection version 2 (Brodzik et al. 2012, 2014) with a nominal 36 km x 36 km pixel size (Chan et al. 2016). Enhanced radiometer-based data products are available with a pixel size of 9 km x 9 km, which are created through application of the Backus-Gilbert optimal interpolation technique to the antenna temperatures of overlapping SMAP footprints (Chan et al. 2018).

Soil moisture retrieval is performed using five algorithms applied to the calibrated and gridded L1C brightness temperature products, and ran through sets of processing steps and data streams responsible for the necessary inputs and flags for unfavourable land surface conditions (see O'Neill et al. (2018) for details). All five algorithms are based on the zero[th] order radiative transfer approach described in Mo et al. (1982), whereby the most fundamental difference among them is the manner in which the vegetation effects are treated. Two Single Channel Algorithms (SCA) adopt the ancillary data approach (Jackson 1993) to calculate the vegetation opacity and estimate the soil moisture content from either the H or V polarized $T_b$, hereafter referred to as SCA-H and SCA-V. The other three algorithms make use of the combination of H and V polarized $T_b$'s to derive simultaneously the vegetation opacity and soil moisture content. The Dual Channel Algorithm (DCA) and Extended DCA (E-DCA) accomplish this by iteratively minimizing cost functions composed of the observed and modelled $T_b$, whereas the Microwave Polarization Ratio Algorithm (MPRA, Owe et al. 2008) uses closed-form relationships. Previous analyses (Chan et al. 2016) have shown that the E-DCA and MPRA provide essentially the same results as the DCA; therefore, only the retrievals obtained using the SCA-H, SCA-V and DCA are evaluated here.

To optimize the number of covered soil moisture monitoring stations, the SMAP team developed a validation grid processing procedure that allows centring the footprint on any location along the defined 3 km grid lines (Colliander et al. 2017). Figure 2 shows the three SMAP reference pixels selected for the Twente network, coded SMAP 3606, SMAP 3306 and SMAP 4371. SMAP 3606 is the reference pixel based on the native 36 km resolution. SMAP 3306 with a 33 km resolution originates from the development of the enhanced data products (Colliander et al. 2018). SMAP 4371 evolved from a reconsideration of the shape of footprint and has a 36 km resolution. As indicated in Table 1 and shown in Fig. 2, SMAP 3606 and 3306 are quite similar and are both represented by thirteen monitoring stations, while SMAP pixel 4371 is represented by a total of seventeen stations. Note that stations less than five kilometres from the periphery are also assigned to that reference pixel and that not all stations were continuously operational during the study period. For this research, the SMAP soil moisture retrievals (version R164020) for the three reference pixels have been analysed from April 2015 till December 2018.

**3.2. LHM soil moisture simulations**

The LHM (Landelijk Hydrologisch Model) is the national implementation of the Netherlands Hydrological Instrument (NHI, De Lange et al. 2014), which is an integrated framework that couples physically-based modelling approaches for the





groundwater, unsaturated and surface water flow. MODFLOW (Harbaugh et al. 2017) is used to simulate groundwater flow and a meta version of the Soil Water Atmosphere Plant (SWAP, Van Dam et al. 2008) model, called MetaSWAP (Van Walsum and Groenendijk 2008), is adopted to compute the unsaturated soil water flow with a quasi-steady state solution of Richards' equation. Readers are referred to De Lange et al. (2014) and www.nhi.nu (last access: July 10th, 2019) for more information

on NHI.

The national NHI implementation, viz. LHM, consists of boundary conditions and atmospheric forcings derived from a comprehensive set of data layers for which various research institutes in the Netherlands are responsible. For instance, subsurface information follows from the national hydrogeological database with a pixel spacing of 100 m (https://www.dinoloket.nl/regis-ii-het-hydrogeologische-model last access: July 10th, 2019). The Dutch class pedotransfer

function applied to the 500 m resolution soil map provides the soil physical characteristics (Wösten et al. 2013). A 5 m resolution LIDAR-based Digital Terrain Model is included as elevation data (http://www.ahn.nl/index.html, last access: July 10th, 2019) and the land use information is adopted from the national 25 m resolution land use map (Hazeu et al. 2014). Daily rainfall and $ET_{ref}$ are the two primary atmospheric forcings obtained from KNMI as 1 km gridded data products. The precipitation data is based on gauge-adjusted rain radar observations (Overeem et al. 2009) and the spatial $ET_{ref}$ data is obtained

via thin plate spline interpolation of the $ET_{ref}$ available for the thirty-five automated weather stations operated by KNMI in the Netherlands (https://data.knmi.nl/datasets/EV24/2, last access: July 10th, 2019), of which three are shown in Fig. 2.

The LHM simulations used for this investigation cover the period from January 2015 till October 2018. The initial conditions are obtained by spinning the model up for one year starting from long term climatology states. The LHM model structure provides daily soil moisture simulations of the root zone with a nominal depth of 0.4 m for the Twente region on a 250 m

resolution grid.

## 4. From point to footprint scale

### 4.1. Method

Soil moisture measured *in situ* is only representative of a small domain whereas model simulations provide spatial distributions across scales. The assumption is that the spatial soil moisture variability simulated by LHM projected on the *in situ*

measurements can be used to obtain a reference representative for the satellite footprint. Figure 3 illustrates this spatial-scale mismatch between the *in situ* monitoring stations, the LHM grid and the reference pixels of SMAP.

However, spatial-scale mismatch is not the only cause of biases between the climatologies of *in situ* measurements and model simulations. Also, the model physics employed and the specific application (e.g. atmospheric forcings, boundary conditions, soil and vegetation parameters) contribute. The method selected for the transformation from point to footprint scale is based

on matching the mean and standard deviation (1st and 2nd statistical moments) of the *in situ* measured and LHM simulated soil moisture records as was done previously by, for instance, Draper et al. (2009) for the assessment of satellite derived soil moisture over Australia.





The first step is to convert the statistical moments of the LHM simulations into those of *in situ* measurements within the satellite footprint, according to,

$$\theta_{m,\mathrm{f}}^{i}(t) = \mu_i + \frac{\sigma_i}{\sigma_{m,\mathrm{p}}}\left[\theta_{m,f}(t) - \mu_{m,\mathrm{p}}\right] \tag{1}$$

where $\theta$ (m$^3$ m$^{-3}$) stands for the spatial mean soil moisture content within the SMAP footprint of the indicated data source at

time $t$, $\mu$ (m$^3$ m$^{-3}$)and $\sigma$ (m$^3$ m$^{-3}$) stand for the temporal mean and standard deviation of $\theta(t)$, subscripts $_i$, $_{m,\mathrm{p}}$ and $_{m,\mathrm{f}}$ indicate that the variable is derived from the *in situ* measurements, model grid cells where the monitoring stations are located, and all model grid cells covering the SMAP footprint, respectively, and superscript $^i$ indicates that the variable is transformed into the *in situ* measured climatology.

After transformation of the LHM soil moisture data to the *in situ* measured statistics, the step from point-scale to footprint

representative measurements is taken as follows,

$$\theta_{i,f}(t) = \mu_{m,f}^{i} + \frac{\sigma_{m,\mathrm{f}}^{i}}{\sigma_i}(\theta_i(t) - \mu_i) \tag{2}$$

where subscript $_{i,f}$ indicates that the *in situ* soil moisture is considered representative for the satellite footprint.

From Eq. (2) a linear relationship between $\theta_{i,f}$ and $\theta_i$ can be identified of the following form,

$$\theta_{i,f}(t) = \mathrm{a} + \mathrm{b}\theta_i(t) \tag{3}$$

with,

$$a = \mu_{m,f}^{i} - \frac{\sigma_{m,f}^{i}}{\sigma_i}\mu_i \tag{3a}$$

$$b = \frac{\sigma_{m,f}^{i}}{\sigma_i} \tag{3b}$$

The parameterization of *a* and *b* can be simplified by recognizing that $\mu_{m,f}^{i}$ and $\sigma_{m,f}^{i}$ can be obtained from Eq. (1) as follows,

$$\mu_{m,f}^{i} = \mu_i + \frac{\sigma_i}{\sigma_{m,p}}\left(\mu_{m,f} - \mu_{m,p}\right) \tag{4a}$$

$$\sigma_{m,f}^{i} = \frac{\sigma_i}{\sigma_{m,p}}\sigma_{m,f} \tag{4b}$$

Substitution of Eq (4b) into Eq. (3b) gives

$$b = \frac{\sigma_{m,f}^{i}}{\sigma_i} = \frac{\frac{\sigma_i}{\sigma_{m,p}}\sigma_{m,f}}{\sigma_i} = \frac{\sigma_{m,f}}{\sigma_{m,p}} \tag{5}$$

Using Eq. (5) together with Eq. (4a) to eliminate $\mu_{m,f}^{i}$ and $\sigma_{m,f}^{i}$ from Eq. (3a) yields after some rearrangements,

$$a = \mu_i\left(1 - \frac{\sigma_{m,f}}{\sigma_{m,p}}\right) + \frac{\sigma_i}{\sigma_{m,p}}\left(\mu_{m,f} - \mu_{m,p}\right) \tag{6}$$

Equation (3) forms the upscaling function with *a* and *b* given by Eq. (6) and Eq. (5), respectively. The advantage of the mathematical rearrangements leading to Eq. (5) and Eq. (6) is that upscaling functions can be determined directly from the statistical moments (1$^{st}$ and 2$^{nd}$) of *in situ* measured and model simulated soil moisture. Interesting to note is that the slope of the upscaling function, *b*, is independent of the *in situ* measurements. This implicates that the choice on the *in situ* measurements affects primarily the offset, *a*, of the upscaling function as will be shown in the following sections.





## 4.2. Comparison of spatial soil moisture means

The application of the method described by Eqs. (1)- (3) depends on the availability of data from the individual measurement stations. As indicated in Sect. 2.2, the data record of the Twente soil moisture monitoring network has several gaps and also the three SMAP footprints cover different numbers of stations. Table 1 indicates for the stations present within the three SMAP footprints the availability and continuity (e.g. changes in the location of stations) of the 5 cm soil moisture data. The table shows that for each of the three footprints there are seven stations available with a fairly complete record and that after the Spring 2016 the data availability improved considerably.

Ideally, satellite-based soil moisture estimates are validated against a reference established based on as many spatially distributed samples as possible over a time period as long as possible. To investigate whether both criteria can be met, we assessed the agreement between pairs of spatial means composed of measurements from i) the seven stations (hereafter 7-station mean) with the most complete records since SMAP operations began and ii) all stations (hereafter all-station mean) within the respective footprints, available as a complete set for a shorter duration. Figure 4 shows for footprints 3306 and 3606 (hereafter 3306 & 3606) the available spatial means for the period October 2016 to October 2018. The plot illustrates an obvious similarity between the soil moisture time series even though they are determined from different sets of soil moisture stations. However, biases exist as a result of spatially variable hydro-meteorological conditions, e.g. soil types, groundwater tables, rainfall and evapotranspiration. Hence, an 'unbiased 7-station mean' obtained by matching the mean and standard deviation is also given in Fig. 4.

Table 2 provides the error metrics computed between pairs of the 7-station (biased and unbiased) and all-station means for footprints 3606 & 3306 and 4371. In general, the metrics confirm the visual interpretation of Fig. 4. A very high correlation, $r$ > 0.99, was found between the spatial means and the mismatch is largely determined by a systematic bias. For instance, for footprint 4371, a RMSE of 0.055 $m^3$ $m^{-3}$ is obtained and the bias is 0.054 $m^3$ $m^{-3}$. After removal of the biases, as was done for the unbiased 7-station, the RMSE (RMSE-b) reduces to 0.009 $m^3$ $m^{-3}$ for footprint 4371 and to 0.013 $m^3$ $m^{-3}$ for footprint 3306 & 3606. From these low RMSE-b values it can be concluded that with the linear equation, the 7-station mean can be used to mimic the all-station mean with a relatively small uncertainty. In the remainder of the manuscript, we utilize the unbiased 7-station mean as one of the validation references.

Further, it should be noted that more than twice as many matchups are available for footprint 3306 & 3606 (592 #) as compared to 4371 (262 #). This is explained by changes in the location of the stations (e.g. 19 and 6) within footprint 4371, which compromises the integrity of spatial mean soil moisture and making it unsuitable for the analysis presented in this section.





## 5. Development of upscaling functions

### 5.1. Measurements versus simulations

In this section we evaluate the agreement between the spatial soil moisture means derived from the *in situ* measurements and the LHM simulations for the SMAP footprints. Figure 5 presents results for January 2015 till October 2018 the LHM simulated
and *in situ* measured spatial mean for footprint 3306 & 3606, whereby the LHM soil moisture is the mean of the simulations for the grid cells where the monitoring stations are located. The two references based on the *in situ* measurements are i) the all-station mean, and ii) the unbiased 7-station mean introduced in Sect. 4.2.

In general, Fig. 5 illustrates that the measured seasonal cycle of wet winters (November to mid-March) and dry summers (mid-May to mid-October) is well reproduced by the LHM simulations. Also, the shorter time scale wetting and drying events
measured at the monitoring stations coincide with the simulated events. As such, it may be concluded that the LHM root zone simulations capture the dynamics of soil moisture measured at a 5 cm soil depth.

However, systematic discrepancies can be noted as well. For instance, the measured winter soil moisture is typically higher than the LHM simulations, while this is the opposite for summers. This may be attributed to the difference in soil layer thickness for which information is provided. In the case of the *in situ* measurements, the probes have an 4 cm influence zone
(e.g. Benninga et al. 2018) and, thus, provide information for the 1 - 9 cm soil layer, while the LHM root zone layer has for the Twente region a nominal depth of 40 cm. This explains the overestimation by LHM in the summer and partly the underestimation in the winter as the larger soil reservoir takes longer to fill up/to deplete, and has also more direct interactions with the groundwater. The winter underestimation is also caused by the fact that in reality the saturated soil moisture content near the surface is higher than prescribed in LHM due to an elevated organic matter content and standing water that may occur.
Exceptions of the LHM winter underestimation occur when the soil is frozen. Under those circumstances, the water in the soil becomes ice and the dielectric constant of the soil mixture measured by the probes drops considerably, while the simulated soil moisture remains unaffected; a good example is February 2018. An exception to the summer overestimation by LHM is found for 2018, which was such a dry period that forced depletion of the LHM root zone to moisture levels similar to those measured at a 5 cm soil depth.

Comparable observations can be made for the LHM simulated and *in situ* measured spatial means of footprint 4371, but a time series plot is omitted here for brevity. Instead Fig. 6 presents scatter plots with the LHM simulated spatial mean of footprint 4371 against the *in situ* measured a) unbiased 7-station and b) all-station mean, in which the matchups for individual years are separated. Two additional scatter plots (c and d) are provided for which the mean and standard deviation of the LHM simulations are matched to those of the *in situ* measured spatial means. The error metrics, *r*, RMSE and bias, computed between
the matchups are given in the lower right corner of the individual plots.

The scatter plots show a linear relationship between the LHM simulated root zone soil moisture and the measured references for the 5 cm soil depth over the range from 0.2 $m^3$ $m^{-3}$ up to 0.7 $m^3$ $m^{-3}$. The discontinuity at 0.2 $m^3$ $m^{-3}$ can be attributed to the summer matchups, in particular those from 2018. Under normal circumstances, the 5 cm soil moisture decreases quite fast





early in the summer after which further drying is constrained by the available water itself. The root zone soil moisture simulated by LHM remains at a higher level longer partly due to a water supply from the deeper layers fed by the groundwater reservoir. However, during prolonged dry episodes the groundwater table drops to levels where the supply from the deep layers is cutoff leading to an accelerated decrease in the LHM simulated root zone soil moisture.

Carranza et al. (2018) made similar observations based on an analysis of the Twente measurements taken at 5 and 40 cm, and referred to this as the (de)coupling of surface and subsurface soil moisture. An interesting implication of the coupled situation under wet conditions is that satellite observed surface soil moisture, such as the SMAP products, can be used as proxy for the root zone soil moisture. Indeed, Pezij et al. (accepted for publication) adopt this assumption by assimilating the SMAP L3 9-km product into the LHM. Note that this is only justifiable for coupled surface-subsurface conditions, which occurs mostly in

regions with sufficiently shallow groundwater tables.

Regardless of the imperfections in the matchups described above, the agreement found between the simulations and measurements can be considered good, with $r > 0.88$. Yet, the RMSEs computed from matchups with the unbiased 7-station and all-station spatial means are, with 0.084 $m^3$ $m^{-3}$ and 0.061 $m^3$ $m^{-3}$ respectively, somewhat unsatisfactory as it is larger than the accuracy requirement defined for the SMAP mission. Even after matching of the statistical moments of the LHM time

series to those of the *in situ* measured references, the RMSE remains above the SMAP requirement with 0.061 $m^3$ $m^{-3}$ and 0.047 $m^3$ $m^{-3}$, respectively.

## 5.2. Accounting for bimodality when deriving upscaling parameters

The utility of the LHM simulations for development of the upscaling functions depends on the linearity between simulated and *in situ* measured soil moisture. Section 5.1 showed that a linear relationship that worked well for wet conditions that does

not capture the data collected under dry circumstances. This suggests that the density distribution of the soil moisture simulated by LHM has bimodal characteristics. To assess this bimodality, the estimated density distribution of LHM soil moisture was reproduced using the weighed sum of two normal probability density functions (pdfs) formulated as follows,

$$g(x) = \alpha f(x, \mu_1, \sigma_1) + (1-\alpha)f(x, \mu_2, \sigma_2) \tag{7a}$$

with

$$f(x, \mu, \sigma) = \frac{1}{\sigma\sqrt{2\pi}} e^{\left(-\frac{(x-\mu)^2}{2\sigma^2}\right)} \tag{7b}$$

where $x$ is taken here as $\theta_{m,p}$, $\alpha$ is a weighing parameter and subscripts 1 and 2 represent the statistical moments of pdf 1 and pdf 2. Equation (7) is matched to the density distribution (bin size 0.01 $m^3$ $m^{-3}$) estimated from the LHM soil moisture simulations by fitting the parameters $\alpha$, $\mu_1$, $\mu_2$, $\sigma_1$ and $\sigma_2$. The $r$ is selected as the objective function, which is optimized using the Generalized Reduced Gradient (GRG, Lasdon et al. 1978) method implemented in the Solver add-in of MS Excel. Figure

7 shows the density estimated from the LHM soil moisture simulations and calculated using the weighed sum of two normal distributed pdfs (hereafter bimodal pdf) with the individual contributions of the two pdfs presented as well. Table 3 lists the fitting parameters for footprints 3606 & 3306 and 4371; both are obtained with a $r$ larger than 0.989.



Figure 7 illustrates that the bimodal pdf matches the density of the LHM soil moisture simulations reasonably well, which is also supported by the large $r$. The underlying fitted pdfs include a pdf (pdf 1) with a $\mu$ larger than 0.32 m$^3$ m$^{-3}$ and a small $\sigma$ of 0.036 m$^3$ m$^{-3}$, and a pdf (pdf 2) with a $\mu$ smaller than 0.24 m$^3$ m$^{-3}$ and a $\sigma$ larger than 0.057 m$^3$ m$^{-3}$. Even though attribution is difficult, pdf 1 may be interpreted as the winter situation during which the dynamic soil moisture range is constrained by

shallow groundwater tables on the down (dry) side and porosity on the up (wet) side. Hence, pdf 2 may be considered as the summer situation.

With the established modes of the density of LHM soil moisture simulations, the available data can be split into sets that have a normal distribution. Subsequently, Eqs. (5) and (6) can be applied to arrive at parameters, $a$ and $b$, for the upscaling function, Eq. (3). Here, the threshold $> \mu_1 - 2\sigma_1$ taken to extract the set representing pdf 1, which implicates a value of 0.251 m$^3$ m$^{-3}$

for footprint 3606 & 3306 and 0.259 m$^3$ m$^{-3}$ for footprint 4371. Table 4 lists the scaling parameters calculated using the pdf 1 set with the unbiased 7-station and the all-station mean as reference. In addition, the parameters calculated with the complete data set are provided. Important to note here is that the slope, $b$, of the upscaling function is independent of the selected *in situ* reference as demonstrated in Sect. 4.1.

## 6. Validation

### 6.1 Time series

Figure 8 shows time series of the SMAP estimates of a) SCA-H, b) SCA-V and c) DCA, for reference pixel 3606 and the native all-station mean *in situ* soil moisture for the complete study period regardless of gaps in the time series of *in situ* measurements, along with the mean daily minimum air temperature and daily rainfall measured at the three KNMI weather stations. The results for the SMAP footprints 3306 and 4371 are included in the supplement in Figs. S1 and S2, respectively.

In general, the time series of the SMAP and *in situ* soil moisture display a great similarity on a seasonal scale, particularly for the SCA estimates. Both the SMAP and *in situ* soil moisture are high during winters mainly due to a weak evaporative demand, see Fig. 1. This evaporative demand is much stronger during summers resulting in lower SMAP and *in situ* soil moisture levels. In the SCA series the dry-down events can be recognized, which is less the case for the DCA record. Overall, the DCA estimates have a larger apparent volatility than the SCA soil moisture and the SCA-H soil moisture reaches its upper limit under wet conditions more frequent than the SCA-V estimates.

For all three algorithms, large mismatches between the SMAP estimates and *in situ* measurements manifest themselves as i) overestimations after rain events demarcating the end of dry spells, and ii) underestimations during winters. Examples of the former are obvious in the fall of 2016 and in August 2018. The winter underestimations coincide with sub-zero temperatures that are not masked via the SMAP freeze/thaw flag that is derived from global land model simulations and the frost depth is

too shallow to also affect the *in situ* sensors. On the other hand, there is a reasonable agreement between the SMAP estimates and *in situ* measurements when frost in the soil does reach the sensors, see for instance February/March 2018.



## 6.2 Matchups

A quantitative assessment of the three SMAP products is made for matchups for which the data from all stations contributing to the unbiased 7-station mean are available. These matchups were used to compute the *r*, bias and uRMSE defined in the appendix and presented in Figs. 9, 10 and 11, respectively. The *r* is provided for the three SMAP reference pixels, for the

morning and afternoon overpasses, and for the unbiased 7-station and all-station mean soil moisture as reference. Here, no differentiation is made between the different implementations of the upscaling method as it does not affect the agreement between matchups due to its linear nature. The upscaling method does influence the biases and uRMSEs, and are, therefore, given for the native *in situ* references and the references whereby the upscaling parameters are derived using i) the full set of LHM simulations (hereafter LHM-all), and ii) the LHM simulations belonging only to the wet regime (hereafter LHM-pdf1).

A selection of scatter plots showing SMAP soil moisture retrievals versus references based on measurements is given in Fig. 12 for reference pixel 4371. The full set of plots is presented as supplement in Figs. S3-S8. In total 601 matchups are available for SMAP's morning overpasses for reference pixel 3606, 621 matchups for 3306, and 666 matchups for 4371. The number for the afternoon passes was 719, 731 and 722 matchups for reference pixels 3606, 3306 and 4371, respectively.

The overall results of the assessment represented by the error metrics and scatter plots are in line with the previous validation

reports by the SMAP validation team (e.g. Chan et al. 2016, Colliander et al. 2018, Chan et al. 2018). The soil moisture retrieved with the SCA-V yields consistently better *r* and uRMSE than the estimates with the SCA-H and DCA. The DCA matchups have a larger spread with an average *r* of 0.764, than found for the SCA-V and SCA-H, with an average *r* of 0.838 and 0.818, respectively. The SCA-H performs slightly less than SCA-V in terms of *r* and this lower performance is comparable to the one found for the DCA in terms of uRMSE. This is attributable to a more frequent saturation of the SCA-H estimates,

which is particularly clear from the scatter plots for the afternoon overpasses, see Fig. 12 *x*–iii and *x*-iv. As the assumption of vertically uniform temperature and dielectric profiles is generally considered to have a higher validity near dawn (O'Neill et al. 2018), it was unexpected that the error metrics are consistently in favour of the afternoon over the morning matchups. On the hand, Jackson et al. (2012) also found a better agreements between afternoon SMOS retrievals and *in situ* measurements. This is further discussed in Sect. 7.2.

With regard to the different implementations of the upscaling strategy, it is clear that the largest impact is found for the bias (Fig. 10) and only a minor effect can be noted across the uRMSEs (Fig. 11). Moreover, the upscaling strategy affects the bias negatively; the biases are larger when the SMAP estimates are assessed against the upscaled *in situ* references. This is partly because the bias of the SMAP soil moisture towards native *in situ* references is already fairly small and the application of the upscaling method apparently only adds uncertainty. An alternative way to interpret this is that the collection of stations within

the Twente network is representing the spatial mean soil moisture observed by SMAP fairly well. This is supported by the small differences in the error metrics computed with the unbiased 7-station and all-station soil moisture means for reference pixels 3306 and 3606. This also applies to a lesser extent for reference pixel 4371, for which the uRMSE computed with the all-station mean as reference is on average more than 0.005 $m^3$ $m^{-3}$ lower than when the unbiased 7-station mean is used. A





possible explanation for this difference in performance is the number of stations present within the reference pixels. For pixels 3306 and 3606, soil moisture data from on average 11.0 stations contribute to the all-station mean, for pixel 4371 this is 13.7. The latter has almost twice as many spatially distributed measurements in comparison to those used for the unbiased 7-station mean.

Despite all efforts to construct a reliable spatial mean and upscale the collection of point measurements towards the domain of the SMAP reference pixel, it is unfortunate to have to conclude that the accuracy requirements of 0.04 $m^3$ $m^{-3}$ (uRMSE) is not met using the *in situ* references derived from the Twente measurements. An important factor contributing to the large uRMSEs is that in Twente soil moisture measurements cover the full dynamic range from saturated to dry soil. This makes it more challenging to meet uRMSE requirement in comparison to regions with a naturally small soil moisture range as is the case in

arid regions for instance. The best performance with the Twente *in situ* reference is achieved when the SCA-V estimates of pixel 4371 are assessed against the native all-station resulting in an uRMSE of 0.059 $m^3$ $m^{-3}$. This is larger than the 0.054 $m^3$ $m^{-3}$ and 0.056 $m^3$ $m^{-3}$ uRMSE reported in Colliander et al. (2017) and Chan et al. (2018) using matchups constructed from the 2015 and 2016 measurements. It should, however, be noted that in this paper we have analysed the period from April 2015 up to December 2018, during which in 2018 record dry conditions were encountered. Under those circumstances the vertical soil

moisture gradients are strong, which may add to the disparity between soil moisture measured in-situ and estimated via SMAP observations. In the discussion section the error structure and the conditions leading to larger uRMSE will be analysed in more detail.

## 7. Discussion

### 7.1 Error distribution

The error metrics reported in Sect. 6.2 indicate that none of the SMAP products fulfills the uRSME target accuracy using the Twente measurements. Yet, the time series of the SMAP estimates and the *in situ* references in Fig. 8 demonstrates an agreement that is up to the level that the effect of rain events on the soil moisture content can be identified. On the other hand, large mismatches between the SMAP estimates and *in situ* reference are encountered on various occasions. Here, we investigate the overall error structure by constructing histograms from the differences between the SMAP estimates and *in situ* references

shown in Fig. 13. Further, theoretical normal pdfs, Eq. (7b), are derived by fitting the $\mu$ and $\sigma$ for each histogram. The histograms and pdfs are only provided for the SCA-V retrievals from pixel 4371, a) morning and b) afternoon, and with the native i) unbiased 7-station and ii) all station mean as reference.

From a comparison of the histograms and pdfs it follows that the tails hold larger densities, whereby the number of overestimations is larger than the number of underestimation. This implies that the relatively large number of outliers inflates

the 'squared root'-based error metric, uRMSE, which is also supported by the lower magnitude of the $\sigma$ fitted to the histograms. For instance, the lowest uRMSE of 0.059 $m^3$ $m^{-3}$ is found for the SCA-V 4371 afternoon estimates against the all-station mean, while the $\sigma$ fitted through histogram is 0.051 $m^3$ $m^{-3}$ for this case.





## 7.2 Mismatch occurrence

Outliers, e.g. large differences between the SMAP estimates and *in situ* references, are an important cause for the larger uRMSEs than SMAP's target accuracy. Patterns in the occurrence of mismatches are highlighted in Fig. 14 via time series of unbiased differences between SMAP SCA-V 4371 estimates and the all-station mean exceeding twice the target accuracy, > 0.08 $m^3$ $m^{-3}$. The results from SMAP's morning and afternoon overpasses are separated in the plot and the daily minimum air temperature ($T_{min}$) and the all-station mean soil moisture are provided as reference for the environmental conditions.

Figure 14 shows three large groups of matchups where SMAP overestimates the *in situ* reference, namely winter 2015/16, fall 2016 and summer/fall 2018. In the latter two, dry spells were ended by a sequence of substantial rain events that exposed the disparity in sampling depth between SMAP and the *in situ* sensors. Indeed, the *in situ* sensors of Twente network installed at a 5 cm soil depth effectively monitor the 1 - 9 cm soil layer below the surface (Benninga et al. 2018), while SMAP's sampling depth is generally considered to be shallower and depends on the soil moisture content (e.g. Colliander et al. 2017, Zheng et al. 2019). After a dry episode has ended, the soil moisture content in the subsurface will be low and increases towards the surface, causing the overestimation by SMAP. Similar findings are reported in Shellito et al. (2016) through quantification of SMAP and *in situ* soil drying.

However, the winter 2015/16 overestimation was not preceded by a drought. In fact, it was quite wet (see Fig. 1) small scale flooding of agricultural parcels across the Twente region. Standing water lowers the L-band emissivity and we expect that this contributed to SMAP's overestimation during the winter of 2015/16.

As suggested in Sect. 6.1, the large underestimations by SMAP can often be associated with frozen conditions. Notably, ice has a dielectric constant comparable to that of dry soil and, therefore, the SMAP estimates decrease as the ice content in the soil increases. *In situ* sensors also quantify the soil moisture content by measuring the dielectric constant and are likewise influenced by the ice content, see February/March 2018. However, the *in situ* sensors provide measurements over a larger soil depth and are only weakly affected when the frost depth is shallow. Hence, also the underestimation by SMAP follows from a difference in sampling depth. The SMAP soil moisture processor includes a freeze/thaw flag that is derived from global surface temperature simulations of the Goddard Earth Observing System model, version 5 (GEOS-5), which is inevitably subject to uncertainties and cannot capture all frost events.

Figure 14 provides an opportunity to discuss the unexpected superior error metrics found for the afternoon over the morning matchups. It is self-evident that the soil moisture estimated from morning SMAP observations are more susceptible to frost and, thus, to underestimation of the *in situ* reference because air and surface temperatures are lower than in the afternoon. However, also the SMAP overestimations in the fall of 2016 and summer/fall of 2018 are more severe for SMAP's morning observations, e.g. compare the tails of the histograms in Figs. 13 a) and b). A possible explanation could be that when the soil surface is wetter than the subsurface, the SMAP soil moisture in the afternoon will be drier due to evaporation and, therefore, closer to *in situ* reference. In any case, the temperature and dielectric constant stratification across the soil-vegetation system





is not the decisive factor that makes morning satellite observations more favourable for microwave radiometry based soil moisture monitoring over Twente.

In an attempt to quantify the negative impact of frozen conditions and antecedent rainfall on SMAP's performance, uRMSEs have been calculated using validation sets excluding matchups for which the $T_{min}$ did not or the daily amount of rainfall did

exceed a certain threshold. Figure 15 presents the uRMSE against the $T_{min}$ or the rainfall threshold for morning (Fig. 15a) and afternoon (Fig. 15b) SMAP SCA-V soil moisture estimates of reference pixel 4371. The figure illustrates that the agreement between SMAP and *in situ* is considerably less on days with rainfall for both morning and afternoon SMAP estimates. Impact of frozen conditions is comparable on the morning SMAP estimates but of much less significance for the afternoon retrievals for the reasons discussed above. When validation sets are filtered for both frost ($T_{min} < 2$ ºC) and rainfall, the uRMSE drops

even further to 0.053 m$^3$ m$^{-3}$ for the morning and 0.043 m$^3$ m$^{-3}$ for the afternoon SMAP estimates (dashed lines in Fig. 15). The latter becomes quite close to the mission's target accuracy.

## 8. Conclusion

In this paper, we report on the validation of Soil Moisture Active/Passive (SMAP) passive-only soil moisture products using *in situ* measurements collected from April 2015 until December 2018 by the Twente monitoring network situated in the east

of the Netherlands and model simulations by the Dutch National Hydrological Model (LHM). The monitoring network consisted of twenty measurement locations during the study period. However, not all stations provided data continuously due to inevitable equipment failures and changes in landownership forcing re-installation of instrumentation. Seven stations that were within the SMAP reference pixels and had fairly complete data records can be identified. This number is insufficient to construct a reliable spatial mean according to criteria defined by the SMAP validation team. Spatially distributed soil moisture

simulations performed by the LHM are employed to translate the sample mean of a collection of point measurements to the domain of the SMAP reference pixels. To this aim, we have derived a set of equations to compute the parameters of a linear scaling function that uses the mean and standard deviation of time series of the *in situ* measured and model simulated spatial soil moisture means. With this approach biases in these statistical moments between the two datasets are accounted for. The upscaling strategy has been applied to an unbiased spatial mean soil moisture computed from the *in situ* measurements

collected by the seven stations with fairly complete records and the spatial mean soil moisture calculated from all available measurements.

We have adopted the native and upscaled spatial mean soil moisture as *in situ* references to assess the SMAP soil moisture estimates obtained with the i) Single Channel Algorithm at Horizontal polarization (SCA-H), ii) Single Channel Algorithm at Vertical polarization (SCA-V) and iii) Dual Channel Algorithm (DCA). In line with previous validation reports, we find that

the SCA-V is the best performing algorithm. SMAP's afternoon soil moisture estimates are systematically in closer agreement with the *in situ* references than the morning estimates. This can be in part attributed to the fact the most severe over- and underestimations are found for the estimates retrieved from SMAP's morning observations. The large overestimation errors





($> 0.08$ m$^3$ m$^{-3}$) typically occur at the end of dry spells when the soil moisture is higher near surface than in the subsurface. Large underestimation errors ($< 0.08$ m$^3$ m$^{-3}$) are noticed in periods with sub-zero air temperatures causing local freezing, which are not all identified by the SMAP freeze/thaw flag that is derived from global land model simulations. Both under- and overestimations essentially follow from the disparity in the soil depth characterized by SMAP and *in situ* as frost and soil

wetting coincide with strong vertical dielectric/soil moisture gradients, and the satellite probes the soil from the surface, while the sensors are installed 5 cm below the surface.

The large over- and underestimations contribute to the fact that SMAP's target accuracy of 0.04 m$^3$ m$^{-3}$ is not achieved using any of the *in situ* references derived from the Twente measurements. The best unbiased root mean squared error (uRMSE) of 0.059 m$^3$ m$^{-3}$ is obtained for the SMAP's SCA-V afternoon estimates assessed against the native all-station mean. This can

reduce to 0.043 m$^3$ m$^{-3}$ when matchups are filtered for frozen conditions and antecedent rainfall. The upscaled *in situ* references do not results in better metrics. Notably, the application of the upscaling strategy has a negative impact on the bias, which is somewhat inevitable due to the already good agreement between the *in situ* measurements and SMAP retrievals.

Despite the fact that the error metrics found in this case study exceed the target accuracy, we are of the opinion that SMAP is a reliable source of soil moisture information for the Twente region. The large mismatches found are caused primarily by

inherent differences in the soil moisture content observed by SMAP and *in situ* that are governed by different processes that control the exchange mass and heat at the land-atmosphere interface. As such, the mismatches must not be considered as errors, but treated as hydro-meteorological information that can be used for improving model simulations and making better informed decisions to facilitate more skillful management of available natural resources.

**Data availability**

The SMAP soil moisture retrieved for the reference pixels, the *in situ* soil moisture measurements of the Twente network, the LHM soil moisture simulations used for the research presented in the articles are available at: *doi to be added*

**Appendix**

Pearson's product moment correlation coefficient (*r*), bias or Mean Error (ME), Root Mean Squared Error (RMSE), and unbiased Root Mean Squared Error (uRMSE) are the metrics used for the assessments presented in this manuscript. These

metrics can be expressed as follows,

$$r = \frac{\sum_{i=1}^{n}(y_i - \bar{y})\sum_{i=1}^{n}(x_i - \bar{x})}{\sqrt{\sum_{i=1}^{n}(y_i - \bar{y})^2}\sqrt{\sum_{i=1}^{n}(x_i - \bar{x})^2}} \tag{A1}$$

$$bias = \text{ME} = \frac{1}{n}\sum_{i=1}^{n}(y_i - x_i) \tag{A2}$$

$$RMSE = \sqrt{\frac{1}{n}\sum_{i=1}^{n}(y_i - x_i)} \tag{A3}$$





$$uRMSE = \sqrt{\frac{1}{n}\sum_{i=1}^{n}\left((y_i - ME) - x_i\right)} \tag{A4}$$

where $x$ stands for the reference, y for the estimated quantity, $n$ is the total number of samples, $i$ indicate an individual realization and the bar stands for the mean of the respective quantity.

## Supplement

The supplement related to this article is available online at: *doi to be added*

## Author contribution

RV designed the research and wrote the paper with contributions from all other authors. RV and HJB collected the observational data, MP performed the LHM simulations, AC, RB, SC, and TJJ provided the matching SMAP data.

## Competing interests

The authors declare that they have no conflict of interest.

## Acknowledgements

The first author would like to thank dr.ir. Mhd Suhyb Salama and Prof. Alfred Stein for scientific discussions related to research presented in this article and acknowledge the support by the Netherlands Organization for Scientific Research (NWO) via the Optimizing Water Availability with Sentinel-1 Satellites (OWAS1S) project number 13871. The Royal Netherlands
Meteorological Institute (KNMI) is acknowledged for making available meteorological measured collected by their network of automated weather stations. A partial contribution to this work was made at Jet Propulsion Laboratory, California Institute of Technology under contract with the National Aeronautics and Space Administration.

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



**List of tables**

**Table 1.** Data availability of the Twente monitoring stations within SMAP footprints. Q1, Q2, Q3, and Q4 refer to quartiles 1 (January-March), 2 (April-June), 3 (July-September)and 4 (October-December). '√' indicates that the station is used for determining the reference for the SMAP footprint, '*x*' indicates that the data is available for that specific period and station, and '- ' means that the two previous qualifications are not applicable. Dark grey shaded cells suggest that the location of the station was changed; exact dates are given in Table S1. Italic and light grey station ID's highlight that records over period 2015-2018 are fairly complete.

**Table 2.** Error metrics computed from matchups of the spatial soil moisture means derived from measurements collected at seven stations and all stations within the respective footprint. *N* stands for the number of matchups, *r* is Pearson product moment correlation coefficient, *a* and *b* are coefficients of a linear function, $y = a + b\,x$, between the two footprint averages derived as given by Eqs. (3a) & (3b), and uRMSE is computed between the unbiased 7-station and all-station means.

**Table 3.** Parameters of the bimodal probability density functions fitted to density distribution of LHM soil moisture simulations of the grid cells where monitoring stations are located. Note that the results for the two footprints are separated because they contain different sets of monitoring stations.

**Table 4.** Upscaling parameters, *a* and *b,* calculated by applying Eqs. (5) and (6), respectively, to the complete data and a set representing pdf 1 with the unbiased 7-station and the all-station mean as references. The pdf 1 set holds 828 and 747 samples for footprints 3606 & 3306 and 4731, respectively and the complete set includes 1146 and 1003 samples, respectively.


**Table 1. Data availability of the Twente monitoring stations within SMAP footprints. Q1, Q2, Q3, and Q4 refer to quartiles 1 (January-March), 2 (April-June), 3 (July-September)and 4 (October-December). '√' indicates that the station is used for determining the reference for the SMAP footprint, 'x' indicates that the data is available for that specific period and station, and '-' means that the two previous qualifications are not applicable. Dark grey shaded cells suggest that the location of the station was changed; exact dates are given in Table S1. Italic and light grey station ID's highlight that records over period 2015-2018 are fairly complete.**

| Station ID | SMAP footprint | | | 2015 | | | | 2016 | | | | 2017 | | | | 2018 | | | |
|---|---|---|---|---|---|---|---|---|---|---|---|---|---|---|---|---|---|---|---|
| | 4371 | 3306 | 3606 | Q1 | Q2 | Q3 | Q4 | Q1 | Q2 | Q3 | Q4 | Q1 | Q2 | Q3 | Q4 | Q1 | Q2 | Q3 | Q4 |
| *ITCSM02* | - | √ | √ | - | × | × | × | × | × | × | × | × | × | × | × | × | × | × | × |
| *ITCSM03* | √ | √ | √ | - | × | × | × | × | × | × | × | × | × | × | × | × | × | × | × |
| *ITCSM04* | √ | √ | √ | × | × | × | × | × | × | × | × | × | × | × | × | × | × | × | × |
| ITCSM05 | - | - | - | × | × | × | × | × | × | × | × | × | × | × | × | × | × | - | × |
| ITCSM06 | - | - | - | - | - | - | - | - | × | × | × | × | × | × | stopped | stopped | | | |
| *ITCSM08* | √ | √ | √ | × | × | × | × | × | × | × | × | × | × | × | × | × | × | × | × |
| *ITCSM09* | √ | √ | √ | × | × | × | × | × | × | × | × | × | × | × | × | × | × | × | × |
| ITCSM10 | √ | √ | √ | × | × | - | × | - | × | × | × | × | × | × | × | × | × | × | × |
| ITCSM11 | √ | √ | √ | × | × | - | × | - | - | - | × | × | × | × | × | × | × | × | × |
| *ITCSM12* | √ | √ | √ | - | × | × | × | × | × | × | × | × | × | × | × | × | × | × | × |
| *ITCSM13* | √ | - | - | × | × | √ | × | × | × | × | × | × | × | × | × | × | × | × | × |
| ITCSM14 | √ | - | - | × | × | × | × | × | × | × | × | × | × | × | × | × | × | × | × |
| ITCSM15 | √ | √ | √ | × | × | - | - | × | × | × | × | × | × | × | × | × | × | × | × |
| *ITCSM16* | √ | √ | √ | × | × | × | × | × | × | × | × | × | × | × | × | × | × | × | × |
| ITCSM17 | √ | √ | √ | - | - | - | × | × | × | × | × | × | × | × | × | × | × | × | × |
| ITCSM18 | √ | - | - | - | - | - | × | × | × | × | × | × | × | × | × | × | × | - | × |
| ITCSM19 | √ | - | - | × | × | × | × | × | × | × | × | × | stopped | | | | | | |
| ITCSM20 | √ | - | - | × | - | - | × | × | × | × | × | stopped | | | | | | | |
| Hupsel | √ | √ | √ | not operational | | | | | | | | not operational | | | | × | × | × | × |
| Twenthe | √ | √ | √ | not operational | | | | | | | | not operational | | | | × | × | - | × |





**Table 2. Error metrics computed from matchups of the spatial soil moisture means derived from measurements collected at seven stations and all stations within the respective footprint. *N* stands for the number of matchups, *r* is Pearson product moment correlation coefficient, *a* and *b* are coefficients of a linear function, *y* = *a* + *b x*, between the two footprint averages derived as given by Eqs. (3a) & (3b), and uRMSE is computed between the unbiased 7-station and all-station means.**

| metric | units | SMAP footprint | |
| --- | --- | --- | --- |
| | | 4371 | 3306 & 3606 |
| N | # | 262 | 592 |
| Bias | $m^3\ m^{-3}$ | -0.054 | -0.012 |
| RMSE | $m^3\ m^{-3}$ | 0.055 | 0.022 |
| *r* | - | 0.994 | 0.994 |
| *a* | $m^3\ m^{-3}$ | 0.031 | 0.035 |
| *b* | - | 1.094 | 0.919 |
| uRMSE | $m^3\ m^{-3}$ | 0.009 | 0.013 |



**Table 3. Parameters of the bimodal probability density functions fitted to density distribution of LHM soil moisture simulations of the grid cells where monitoring stations are located. Note that the results for the two footprints are separated because they contain different sets of monitoring stations.**

| Footprint | $\alpha$ | $\mu_1$ | $\mu_2$ | $\sigma_1$ | $\sigma_2$ |
|:---:|:---:|:---:|:---:|:---:|:---:|
| | - | | m$^3$ m$^{-3}$ | | |
| *3606 & 3306* | 0.493 | 0.323 | 0.240 | 0.036 | 0.065 |
| *4371* | 0.672 | 0.331 | 0.219 | 0.036 | 0.057 |





**Table 4. Upscaling parameters, *a* and *b*, calculated by applying Eqs. (5) and (6), respectively, to the complete data and a set representing pdf 1 with the unbiased 7-station and the all-station mean as references. The pdf 1 set holds 828 and 747 samples for footprints 3606 & 3306 and 4731, respectively and the complete set includes 1146 and 1003 samples, respectively.**

| Footprint | Parameter | pdf 1 set | | complete set | |
|---|---|---|---|---|---|
| | | *Unb. 7-station mean* | *All-station mean* | *Unb. 7-station mean* | *All-station mean* |
| **3606 & 3306** | $a$ $(m^3\ m^{-3})$ | -0.0397 | -0.0387 | -0.0325 | -0.0317 |
| | $b$ (-) | 0.915 | 0.915 | 0.960 | 0.960 |
| **4371** | $a$ $(m^3\ m^{-3})$ | -0.103 | -0.086 | -0.070 | -0.062 |
| | $b$ (-) | 1.082 | 1.082 | 1.062 | 1.062 |





**List of figures**





when matchups are not filtered, the dashed line is the uRMSE achieved when matchups are filtered on both $T_{min}$ (< 2 ºC) and rainfall (0 mm).





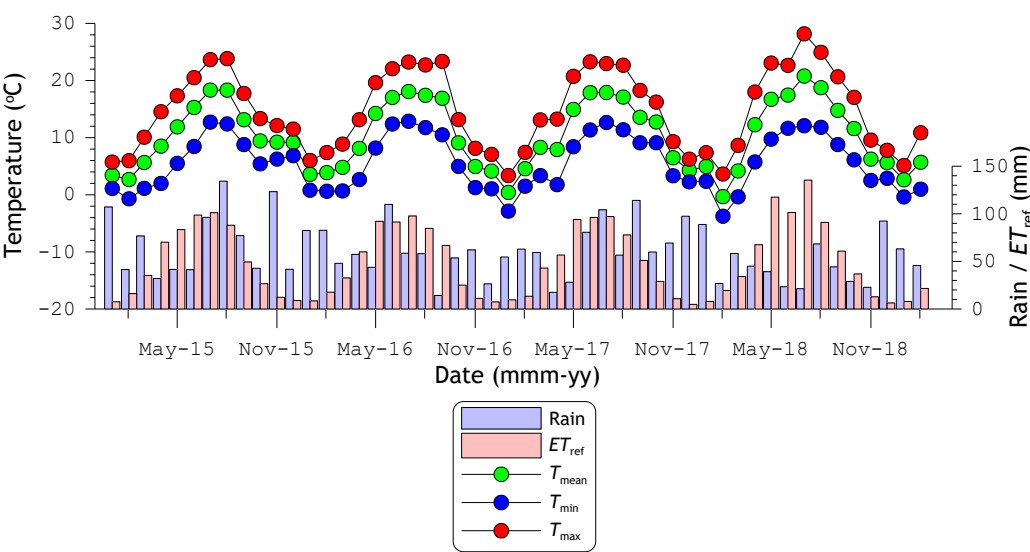

**Figure 1. Monthly averages of daily mean ($T_{mean}$), minimum ($T_{min}$) and maximum ($T_{max}$) air temperature, and monthly sums of rainfall and reference evapotranspiration ($ET_{ref}$) all derived from measurements collected at the three KNMI weather stations Heino, Hupsel and Twenthe.**





**Figure 2. Delineation of the SMAP reference pixels 4371, 3306 and 3606, and the locations of the UT-ITC soil monitoring and the KNMI weather stations lain over a Landsat 8 true colour composite of 20 July 2013 (Image courtesy of the U.S. Geological Survey).** *Double digits* **indicate the soil moisture station ID and** *names* **(Heino, Hupsel, Twenthe) refer to the weather stations. Station with underlined ID numbers are still operational.** *Coordinates* **are given in latitude/longitude (datum: WGS84).**





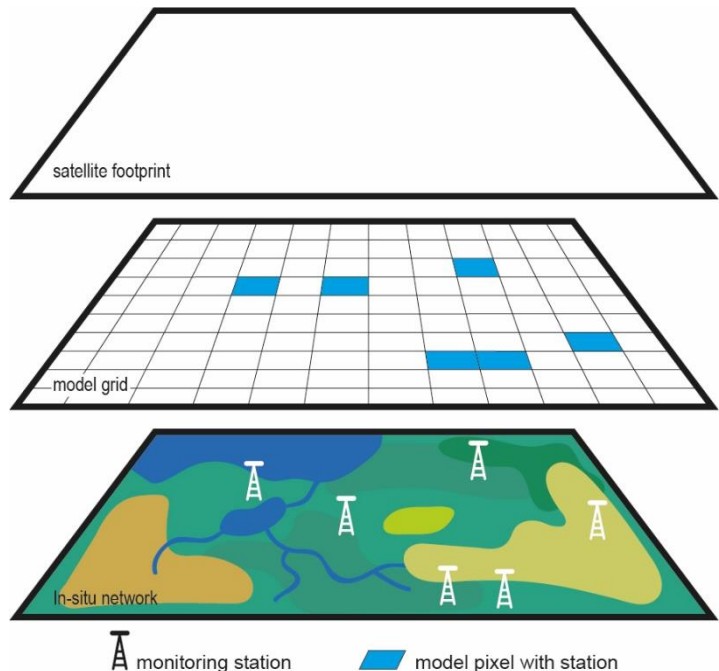

**Figure 3. Schematic illustration of the spatial scale mismatch between** *in situ* **monitoring stations, hydrological model grid and SMAP reference pixel (satellite footprint).**





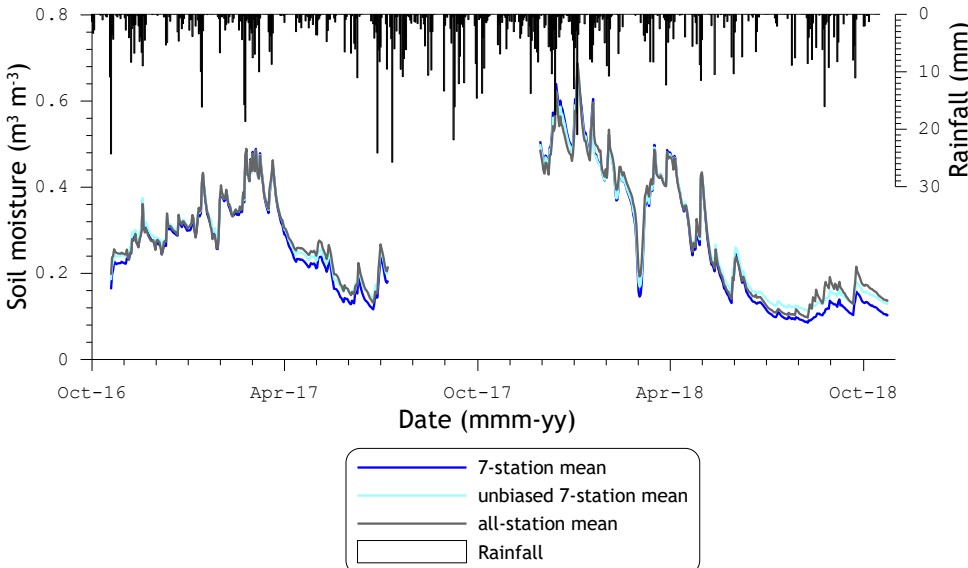

**Figure. 4. Spatial soil moisture means for SMAP reference pixels 3306 & 3606 from October 2016 up to October 2018.**





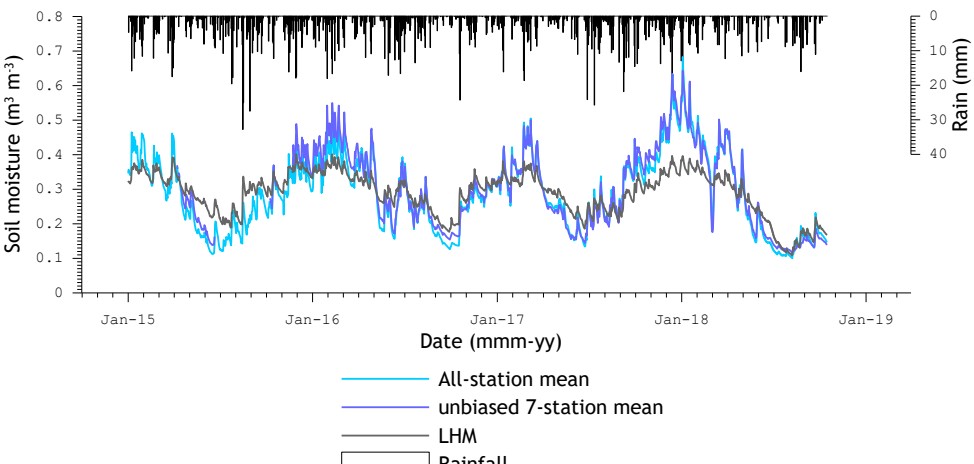

**Figure 5. Spatial soil moisture mean for SMAP reference pixel 3306 & 3606 from January 2015 up to October 2018 derived from i) measurements collected at all available stations, ii) unbiased measurements from the 7 stations operating fairly continuously and iii) LHM simulations for the locations where stations are situated.**

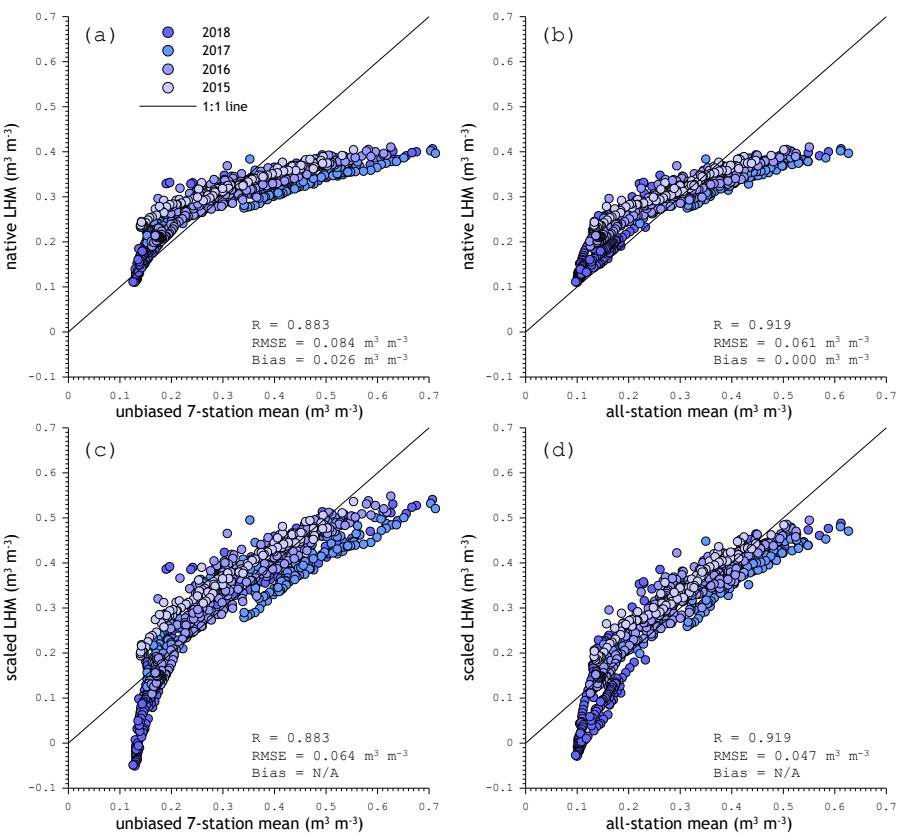

**Figure 6. LHM simulated spatial mean soil moisture against references deduced from *in situ* measurements for SMAP reference pixel 4371; a) and b) show the native LHM simulations versus the unbiased 7-station mean and all-station mean, respectively; c) and d) presented the same as a) and b) only LHM soil moisture is scaled to the statistical moments (1st and 2nd) of the *in situ* references.**





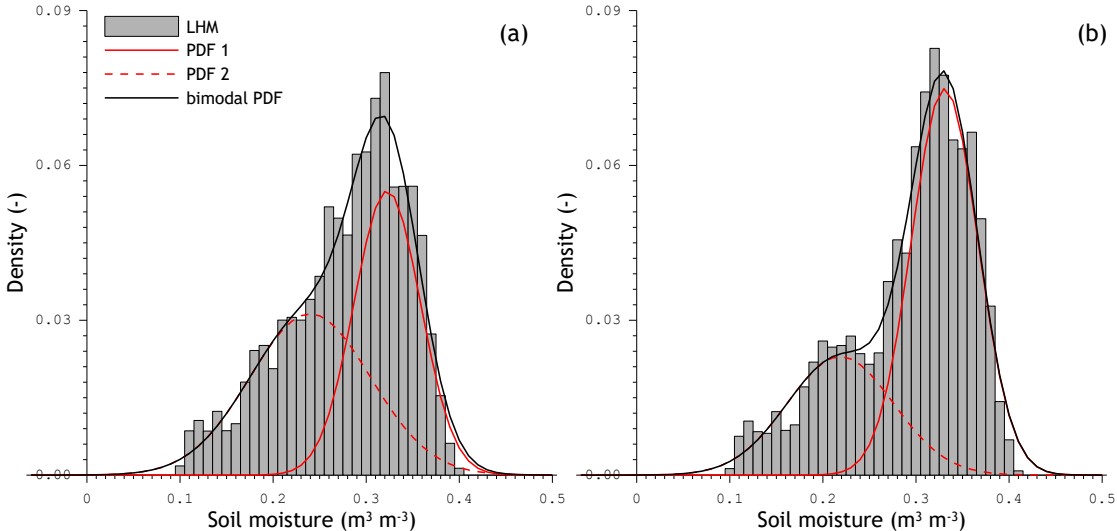

**Figure 7. Density of soil moisture estimated from LHM simulations (bin size 0.01 m³ m⁻³) and calculated using a weighed sum of two normal distributed pdfs. a) shows the density for reference pixels 3306&3606 and b) that of reference pixel 4371.**

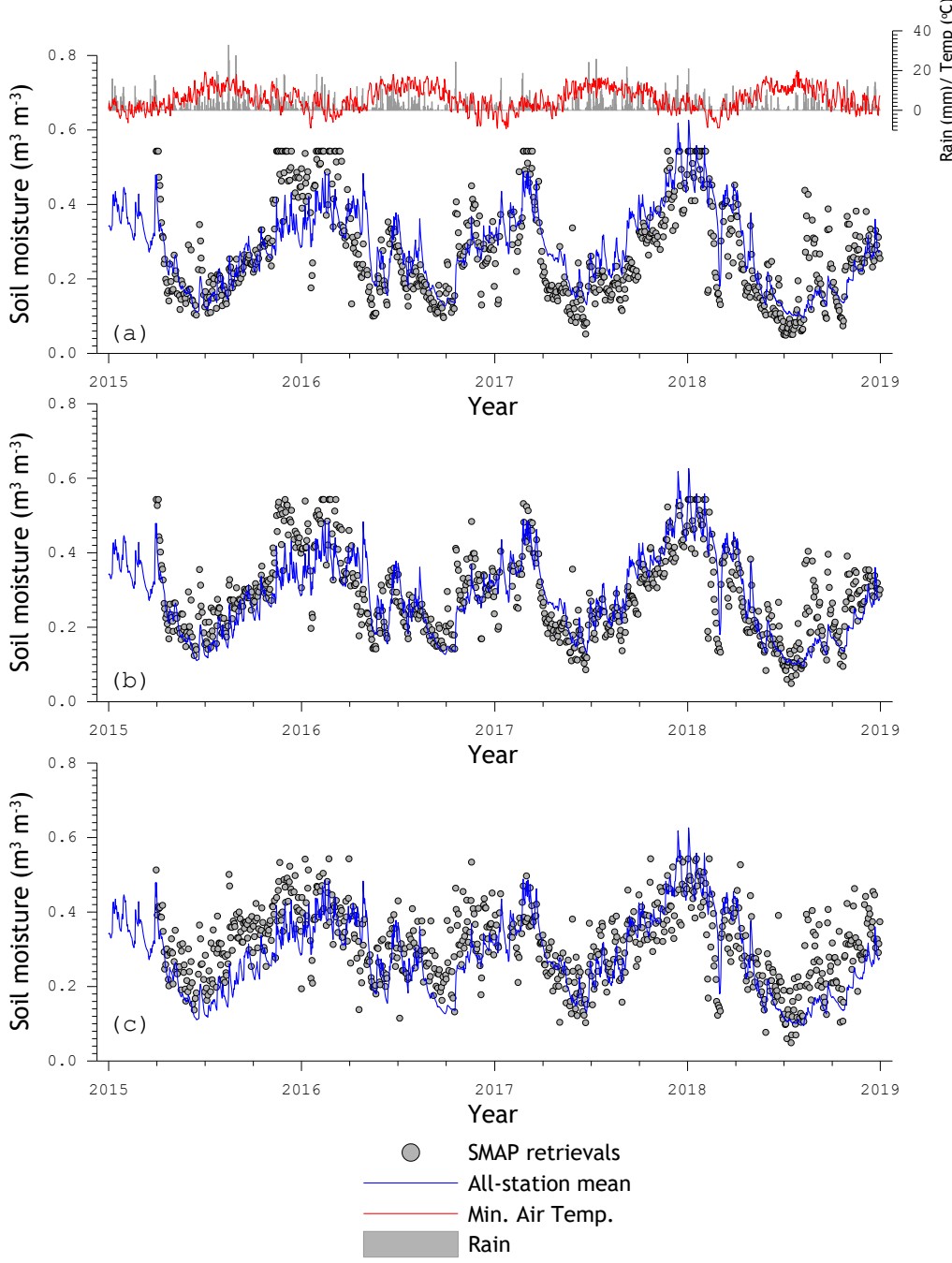

**Figure 8. Time series of daily minimum air temperature, daily rainfall sums, all-station mean soil moisture, and SMAP retrievals for reference pixel 3606 and a) Single Channel Algorithm at H polarization (SCA-H), b) Single Channel Algorithm at V polarization (SCA-V), and c) Dual Channel Algorithm (DCA).**



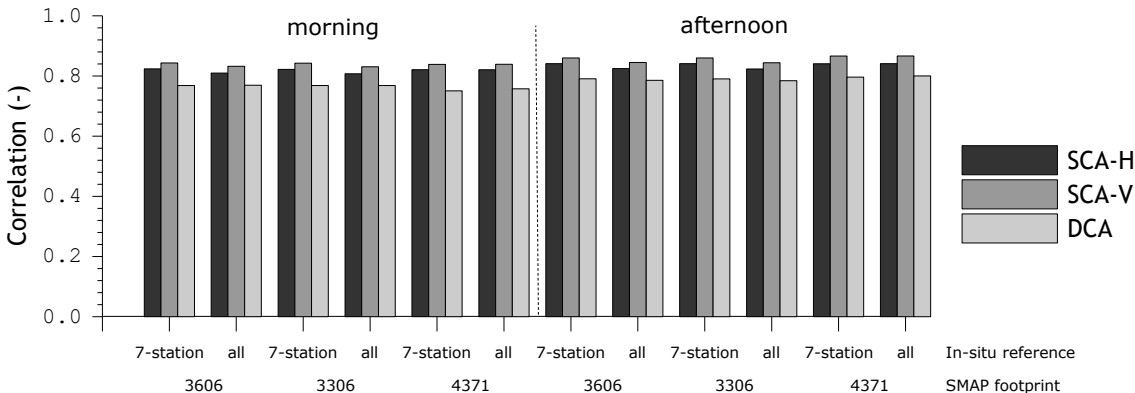

**Figure 9. Pearson product moment correlation coefficient computed between SMAP soil moisture and *in situ* references.**

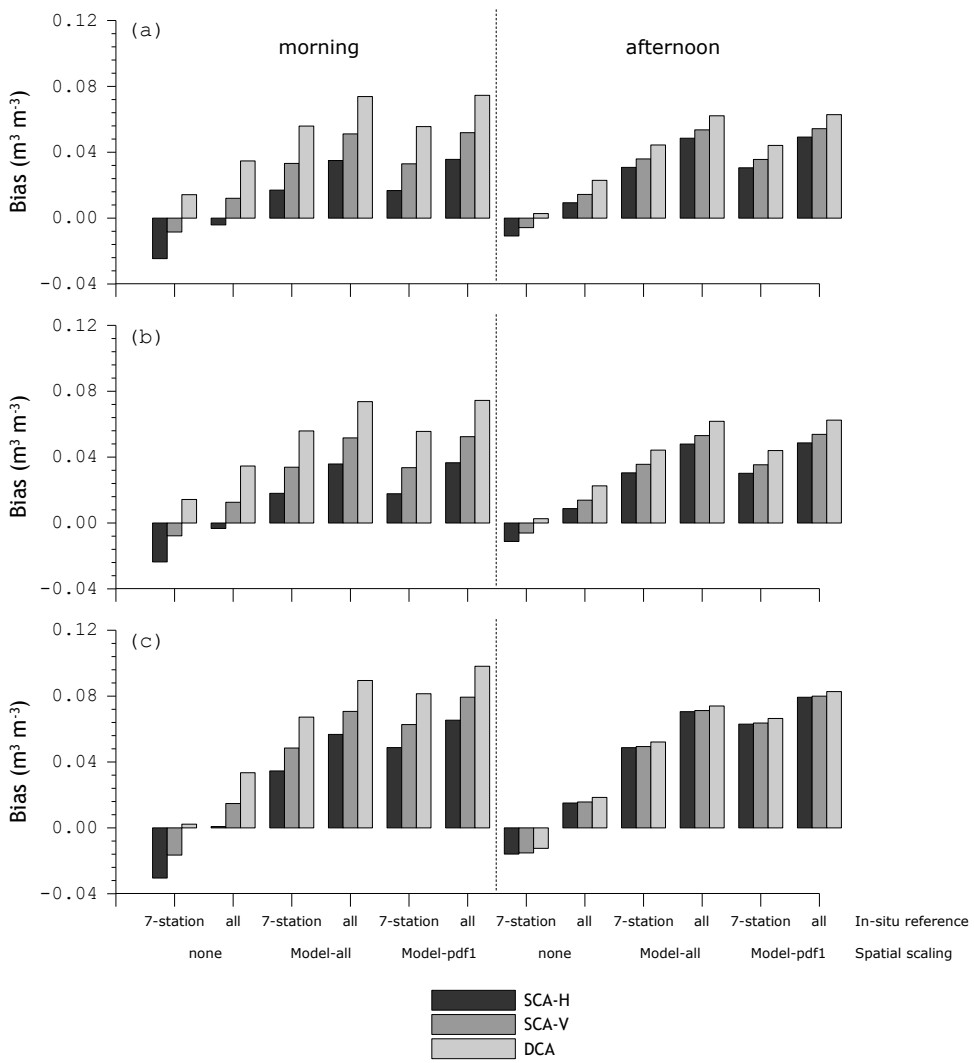

**Figure 10. Bias computed between SMAP soil moisture retrieved and *in situ* references; a) the metrics for reference pixel 3306, b) for reference pixel 3606 and c) for reference pixel 4371.**

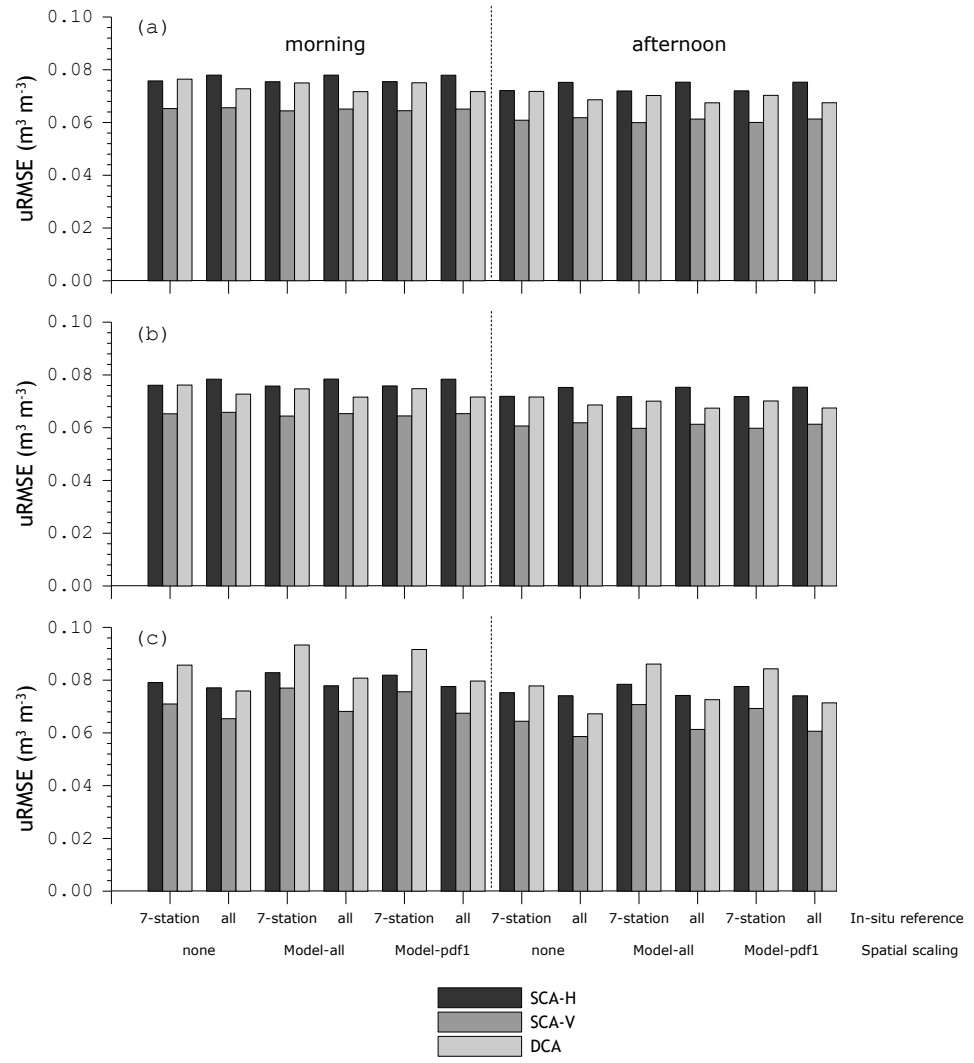

**Figure 11. Unbiased Root Mean Squared Error (uRMSE) computed between SMAP soil moisture retrieved and *in situ* references; a) the metrics for reference pixel 3306, b) for reference pixel 3606 and c) for reference pixel 4371.**





**Figure 12. Scatter plots of SMAP soil moisture retrievals for reference pixel 4371 versus *in situ* references available from April 2015 till December 2018, a), b), and c) indicate SMAP estimates derived using SCA-H, SCA-V and DCA, respectively; i and ii are for SMAP's morning overpass, and iii and iv are for the afternoon overpasses; i and iii include references based on the unbiased 7 station mean soil moisture, and ii and iv use the all-station mean as references.**

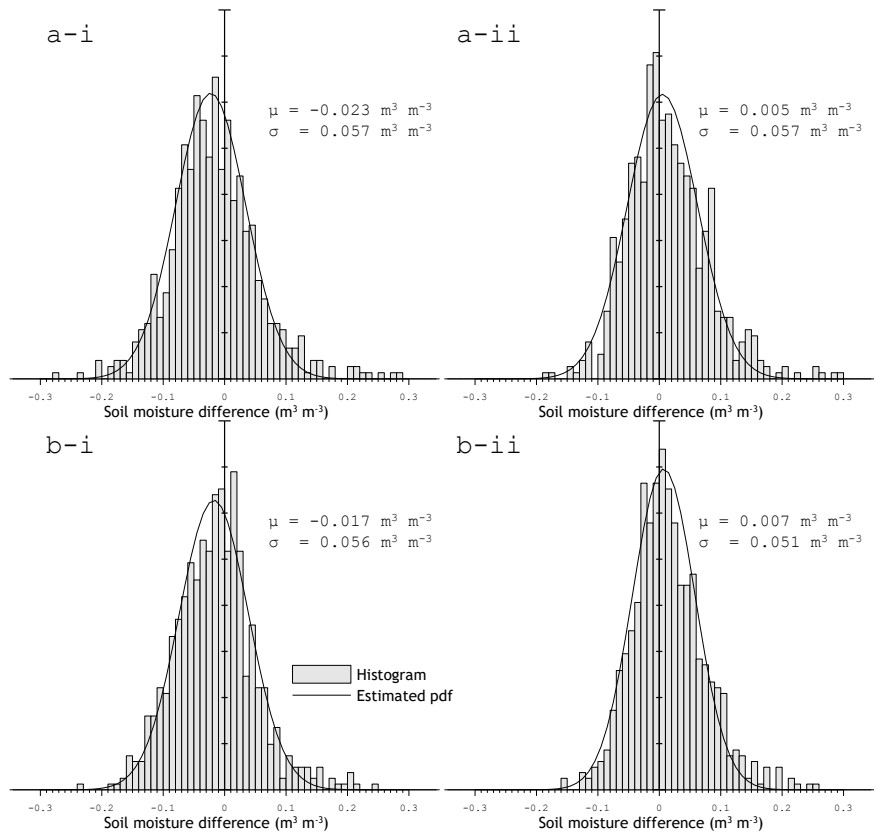

**Figure 13. Histograms constructed from the differences between the SMAP SCA-V retrievals for reference pixel 4371 versus *in situ* references, and probability distribution functions (PDFs) obtained by fitting the mean (*μ*) and standard deviation (*σ*); a) uses the data from SMAP morning overpasses and b) the SMAP afternoon overpasses; i) includes the unbiased 7-station mean as reference, and ii) includes the all-station mean as reference.**



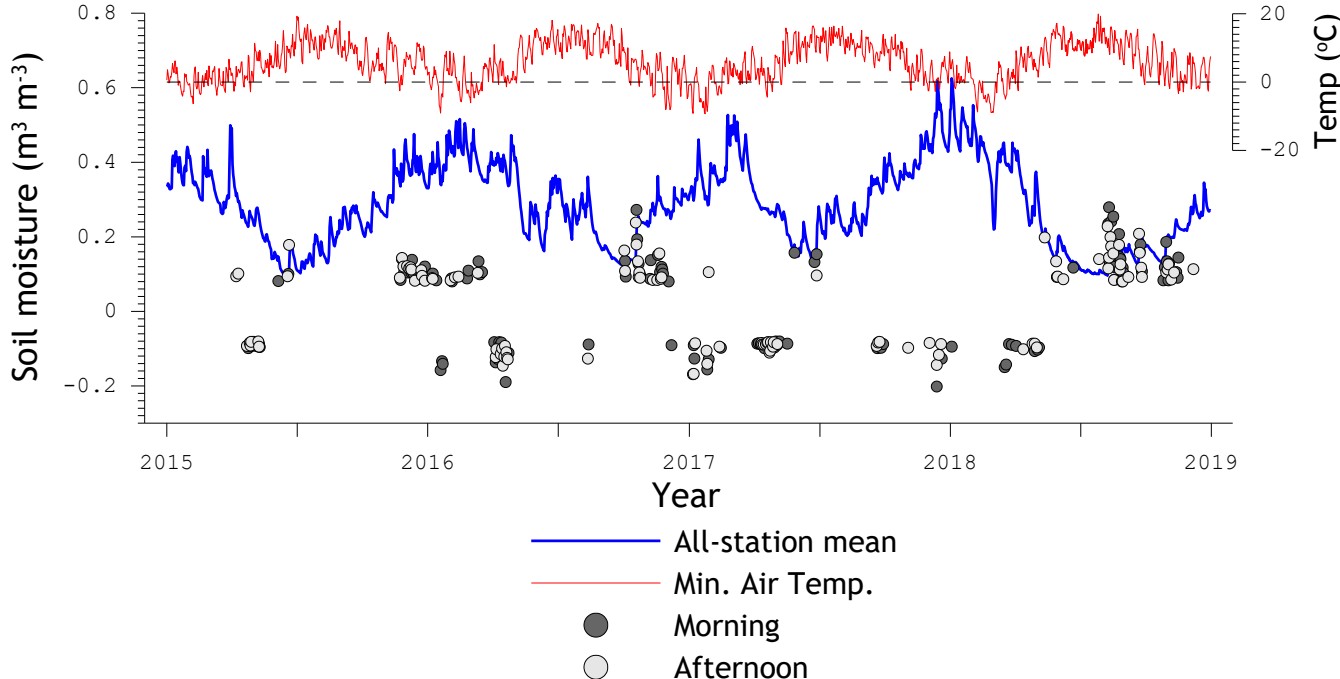

**Figure 14. Time series of measured soil moisture (all-station mean), daily minimum air temperature and unbiased differences between SMAP SCA-V retrievals for reference pixel 4371 and the all-station mean exceeding 2$\sigma$ (>0.08 m³ m⁻³).**



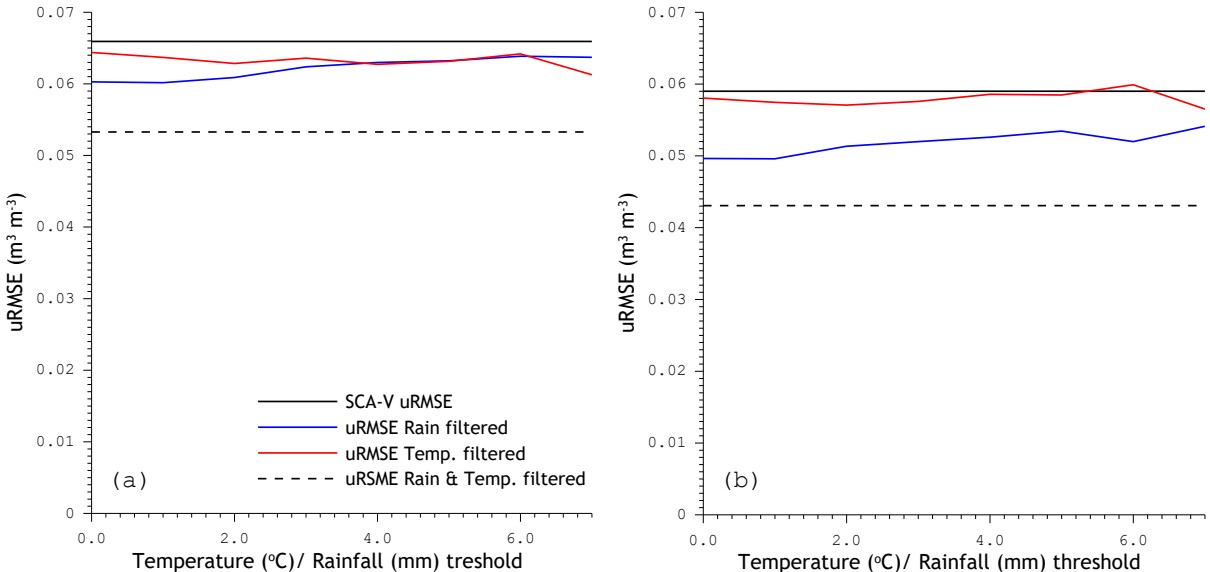

**Figure 15. uRMSEs against the daily minimum 1.5 m air temperature ($T_{min}$) and rainfall threshold used to filter out matchups for suspected frozen conditions and conditions with an disparity between SMAP's sensing depth and the depth at which soil moisture is measured. a) shows the results for the morning estimates and b) the afternoon estimates. The straight line is the uRMSE obtained when matchups are not filtered, the dashed line is the uRMSE achieved when matchups are filtered on both $T_{min}$ (< 2 ºC) and rainfall (0 mm).**