# Peer review of "Validation of SMAP L2 passive-only soil moisture products using upscaled *in situ* measurements collected in Twente, the Netherlands"

_Hydrology and Earth System Sciences, 2019_

## Referee Comment (RC1) · Anonymous Referee #1 · 19 Feb 2020

This manuscript developed a soil moisture dataset using in situ measurements and model simulations and then used the dataset to validate SMAP L2 product. In general, this manuscript is well written and easy to follow. It fits the scope of HESS. However, I have several concerns before considering to publish this work in HESS.

1) Validating satellite soil moisture product is necessary. However, this study focused on a very specific region (one SMAP pixel), which absolutely limits the value of this study. The authors need to clarify how such one-pixel evaluation can advance the understanding of satellite observed soil moisture.

2) If I understand correctly, the upscaling method used in this study is standardizing

the model simulation by in situ observation. I would like to see how much improvement has been made by incorporating in situ values. If the improvement is tiny, then the contribution of in situ data is negligible. It doesn't make sense to assume model simulation as ground truth and to use it to validate other observations.

3) The authors keep using the model simulated root zone soil moisture. Please clarify why you don't use top 5-cm soil moisture from the model.

4) Please describe the uncertainties from in situ measurements and discuss how these uncertainties will influence the findings.

Other specific comments:

1) Title: "the Netherlands". 2) Line 7: "RMSE". 3) Table 2: what does "#" mean? 4) Table 3: reformat the table. 5) Figure 8: enlarge the temperature and precipitation.

---

## Referee Comment (RC2) · Anonymous Referee #2 · 2 Jun 2020

The paper compares SMAP soil moisture estimates with in situ measurements for validation purposes, which is a rather common practice with satellite-derived products. Besides all the technical details and difficulties in the process due to representativeness and scale mismatch, I found that the most interesting (and potentially novel) piece of research consists of the use of a model to get more spatially distributed data and upscale the in situ measurements to the scale of the satellite pixel size. Unfortunately, the upscaling does not seem to produce better results than the raw in situ measurements when compared to the SMAP values. Still, the results are interesting.

My comments are mostly about the model, as I consider that the rest of the methodology is relatively standard. I believe the authors should explain better why they used a model for root-zone soil moisture (40 cm depth) to represent the 5 cm depth soil moisture (both in situ and SMAP estimates). Figures 5 and 6 shows that the match between model and measurements is not very good and I was wondering if the authors tried to get a relationship between the 5 cm and the 40 cm soil moisture values.

The validation is then carried out using the raw in situ measurements because it gives better results. The authors explain mismatches based on physical processes and the inability of SMAP to capture those processes. I would have liked to see more references when describing potential sources of error. The last paragraph does not seem to follow directly from the objectives of the paper or the results presented in the paper. It is also not clear to me how the mismatches can be used to help management.

My last point is regarding the abstract: even though the fact that the upscaling did not work is included in the conclusions, it has not been included in the abstract.

A minor point is that in the supplement, the tittle of the paper wrongly includes the word "upscale".

---

## Author Comment (AC2) · 26 Jun 2020

The authors would like to thank the referee for carefully reading our manuscript and providing constructive criticisms. In our response below we reply to the individual concerns.

Referee 2 comment 1: I believe the authors should explain better why they used a model for root-zone soil moisture (40 cm depth) to represent the 5 cm depth soil moisture (both in situ and SMAP estimates).

Authors' response: The model used in this study is the Dutch national hydrological

model (LHM). This is a modelling framework that couples physically-based modelling approaches for groundwater, unsaturated soil and surface water flow. Particularly the combination of groundwater and unsaturated soil water flow is advantageous for a region with shallow groundwater tables, such as the Netherlands. Another advantage of using LHM is that it makes use of best possible boundary conditions and atmospheric forcings, e.g. 100 m resolution subsurface information, 500 m resolution soil maps and coupled soil physical characteristics, 5 m resolution Digital Terrain Model, 25 m resolution land use map, 1 km resolution daily rainfall and reference evapotranspiration.

We agree with the reviewer that for this research a model with a shallower top soil layer would have been better, but that would have been impossible with the set of boundary conditions and atmospheric forcings available for the LHM. Therefore, we have chosen to use the root zone soil moisture simulated by the LHM and transform this towards the statistical moments of the 5 cm in situ measurements as is presented in section 5.1. We justify the validity of this approach based on the findings of Caranza et al. (2018). They found strong linear relationships between the 5 cm and 40 cm in situ measured soil moisture by the Twente monitoring network suggesting that the surface soil moisture can be used proxy for the root zone soil moisture and vice versa. Pezij et al. (2019) have previously adopted this assumption successfully for the assimilation of the SMAP L3 product into the LHM. Here we used it to develop upscaling functions to translate the spatial mean of point measurements to the domain of the SMAP reference pixels.

Section 5.1 is fully devoted to the justification for using the simulated root zone soil moisture as proxy of the in situ measured soil moisture at 5 cm depth. A discussion on this topic in the context of Carranza et al. (2018) and Pezij et al. (2019) can be found on P10L5-L10. In revised manuscript we will referred to this discussion when we introduce the use of the LHM root zone soil moisture simulations around P6L20. Further it should also be noted that the soil moisture states in process models are defined at the mid-point of the soil layer, 20 cm, and the 5TM probe installed 5 cm

below the soil surface has an approximate 4 cm influence zone and measures over a 1-9 cm soil layer. This information can be partly found on P9L14-L16. We will update the text with the fact that the model states are defined at the mid-point of the root zone layer, e.g. 20 cm.

References: Carranza, C. D. U., van der Ploeg, M. J., and Torfs, P. J. J. F.: Using lagged dependence to identify (de)coupled surface and subsurface soil moisture values, Hydrol. Earth Syst. Sci., 22, 2255-2267, doi:10.5194/hess-22-2255-2018, 2018.

Pezij, M., Augustijn, D.C.M., Hendriks, D.M.D., Weerts, A.H., Hummel, S., van der Velde, R., and Hulscher, S.J.M.H.: State updating of root zone soil moisture estimates of an unsaturated zone metamodel for operational water resources management, J. Hydrol. X, 4, 100040, doi: 10.1016/j.hydroa.2019.100040, 2019

Referee 2 comment 2: Figures 5 and 6 shows that the match between model and measurements is not very good and I was wondering if the authors tried to get a relationship between the 5 cm and the 40 cm soil moisture values.

Authors' response: Indeed the 1:1 match between the measurements and the model simulations has imperfections, but we can also note that both respond in a similar fashion to rainfall inputs as is supported by the high correlation coefficients (> 0.88). In the manuscript in section 5.1 we attribute the difference between the 5 cm in situ measured soil moisture and the root zone soil moisture simulations to discrepancies in soil properties, in soil depth over which information is provided and the decoupling of the surface and subsurface soil moisture when the groundwater retreats under dry conditions.

For this manuscript, we have made use of the complete matchup set as well as the part of the set, for which a linear relationship is found, to develop the upscaling function. We have not included a comparison between the root zone soil moisture simulations and the subsurface in-situ measurements because we do not find this relevant for the objectives of this research.

However, we do find the question interesting and for the response to this referee comment, we have made the comparison between the LHM root zone soil moisture and 20 cm in situ measurements. Note that we have taken the 20 cm measurements because this is at the center of the 40 cm root zone layer in LHM and the model states are defined at the mid-point of the soil layer. The matchups are shown in the figure 1 and the metrics obtained are: a correlation coefficient of 0.95, a RMSE of 0.066 m3 m-3 and a bias of 0.067 m3 m-3. In general, the distribution of the data points in the plot are similar to those in figure 6 except that the spread of the data points is somewhat smaller and discontinuity is less sharp than in the comparison with the in situ soil moisture measured at a 5 cm depth.

Referee 2 comment 3: The validation is then carried out using the raw in situ measurements because it gives better results.

Authors' response: To be precise, we have carried out the validation with two references based on the native spatial mean derived from the in situ measurements and four references based on different versions of the upscaled spatial mean also derived from the in situ measurements. The results showed, however, that the best error metrics are obtained with the reference for which no upscaling has been performed.

In the discussion, section 7, we further analyse the error distribution and hydrometeorological circumstances under which large errors occur. We have chosen to only present the results with the native spatial mean for brevity. This has no influence on the findings in general because the upscaling is linear.

Referee 2 comment 4: The authors explain mismatches based on physical processes and the inability of SMAP to capture those processes. I would have liked to see more references when describing potential sources of error.

Authors' response: In the section 7.2 we refer to Colliander et al. (2017), Zheng et al. (2019) and Shellito et al. (2016) in the context of the dependence of the sampling depth on the soil moisture content. In the revised manuscript we will discuss the impact of

standing water (floods) and frozen conditions also in the context of other investigations. For example, Brakenridge et al. (2007) have demonstrated the use of microwave radiometry for the estimating river discharge based on flood extent and Wegmüller (1990) described the behaviour of microwave signature under frozen and thaw soil conditions. Further, we will refer the SMAP and SMOS literature as the soil moisture products derived from both missions include flags for frozen soils and mitigation measures to account for permanent water bodies.

References: Brakenridge, G. R., Nghiem, S. V., Anderson, E., and Mic, R.: Orbital microwave measurement of river discharge and ice status, Water Resour. Res., 43, W04405, doi: 10.1029/2006WR005238, 2007.

Colliander, A., Jackson, T.J., Bindlish, R., Chan, S., Das, N., Kim, S.B., Cosh, M.H., Dunbar, R.S,. Dang, L., Pashaian, L., Asanuma, J., Aida, K., Berg, A., Rowlandson, T., Bosch, D.D., Caldwell, T., Caylor, K., Goodrich, D.C., Al Jassar, H., Lopez-Baeza, E., Martinez-Fernandez, J., Gonzalez-Zamora, A., Livingston, S., McNairn, H., Pacheco-Vega, A., Moghaddam, M., Montzka, C., Notarnicola, C., Niedrist, G., Pellarin, T., Prueger, J., Pulliainen, J., Rautiainen, K., Garcia-Ramos, J.V., Seyfried, M., Starks, P.J., Su, Z., Zeng, Y., van der Velde, R., Thibeault, M., Dorigo, W.A., Vreugdenhil, J.M., Walker, J.P., Wu, X., Monerris, A., O'Neill, P.E., Entekhabi, D., Njoku, E.G., and Yueh, S.: Validation of SMAP surface soil moisture products with core validation sites, Remote Sens. Environ., 191, 215-231, doi:10.1016/j.rse.2017.01.021, 2017.

Shellito, P.J., Small, E.E., Colliander, A., Bindlish, R., Cosh, M.H., Berg, A.A., Bosch, D.D., Caldwell, T.G., Goodrich, D.C., McNairn, H., Prueger, J.H., Starks, P.J., van der Velde, R. and Walker, J.P.: SMAP soil moisture drying more rapid than observed in situ following rainfall events, Geophys. Res. Lett., 43. 9068-8075, doi: 10.1002/2016/GL069946, 2016.

Wegmüller,U.: The effect of freezing and thawing on the microwave signatures of bare soil, Remote Sens. Environ., 33, 123-135, doi: 10.1016/0034-4257(90)90038-N, 2010.

Zheng, D., Li, X., Wang, X., Wang, Z., Wen, J., van der Velde, R., Schwank, M., and Su, Z.: Sampling depth of L-band radiometer measurements of soil moisture and freeze-thaw dynamics on the Tibetan Plateau. Remote Sens. Environ., 226, 16-25, doi: 10.1016/j.rse.2019.03.029, 2019.

Referee 2 comment 5: The last paragraph does not seem to follow directly from the objectives of the paper or the results presented in the paper. It is also not clear to me how the mismatches can be used to help management.

Authors' response: We agree with the referee that the last paragraph does not follow directly from the results of the paper. We will remove this from the manuscript.

Referee 2 comment 6: My last point is regarding the abstract: even though the fact that the upscaling did not work is included in the conclusions, it has not been included in the abstract.

Authors' response: Our intention was to keep the abstract short with only the most relevant information for the readers. In the abstract we do mention around l16-17 the use of the Dutch national hydrological model for 'the development of upscaling functions to translate the spatial mean of point measurements to the domain of the SMAP reference pixels'. But indeed we do not follow up on the results obtained as we do in the conclusion with 'The upscaled in situ reference do not result in better metrics'. A similar statement we will include in the abstract.

Referee 2 comment 7: A minor point is that in the supplement, the tittle of the paper wrongly includes the word "upscale". Authors' response: Thank you, we will correct this.

———————————————————

[Figure]

Figure 1. LHM root zone soil moisture against *in situ* measured soil moisture at a depth of 20 cm.

**Fig. 1.**

---

## Author Response (AR1)

Dear Prof. Patricia Saco,

Thank you for providing us with the two referee reports. Included in this submission is a marked-up version of the revised manuscript showing the changes made and a document with our point-by-point reply to the referee comments.

The main concerns of *Referee 1* were on the use of a single site for the validation satellite based products and a misunderstanding on the use of model output for the validation. In response to the first comment we would like to argue that the research presented in this manuscript is more than the validation of a soil moisture product. It also deals with the difficulties involved in the creation of consistent references for the assessment of satellite observed soil moisture, such as data gaps in the records of individual measurement locations and spatial mismatch errors through upscaling. For the revised manuscript, we have, therefore, proposed to put more emphasis on the upscaling part by including 'upscaled' in the title. The referee's second comment seems to be based on a misunderstanding that the model simulated root zone soil moisture has been used for the validation of SMAP products. The root zone soil moisture simulation were used to develop upscaling functions. These functions were applied to the 5-cm soil moisture measurements and upscaled in situ measurements used to validate the SMAP soil moisture product. In our response to comments 2 and 3, we have made an attempt to clarify this.

*Referee 2* has made remarks on the use of the root zone soil moisture simulations. In this context, we would like to emphasize that we primarily used the spatially distributed simulations to develop the upscaling functions and the LHM provides the best possible boundary conditions needed for describing the spatial heterogeneity. This we have explained better in the revised manuscript. Further, we present in the manuscript a comparison of 5 cm soil moisture measurements against root zone simulations that is used to unbias the statistical moments of the two data sets. In the response file we also present a 20 cm soil moisture measurements against root zone simulations for the sake of curiosity.

Overall, we believe that with the changes made to the manuscript we have adequately addressed the concerns of the two referee. Therefore, we hope that you will consider it for further review and eventual publication in HESS.

Your truly
Rogier van der Velde
On behalf of the authors

**Anonymous Referee 1**

This manuscript developed a soil moisture dataset using in situ measurements and model simulations and then used the dataset to validate SMAP L2 product. In general, this manuscript is well written and easy to follow. It fits the scope of HESS. However, I have several concerns before considering to publish this work in HESS.

Authors' response:
The authors would like to thank the referee for carefully reading our manuscript and providing constructive criticisms. In our responses below we address the expressed concerns.

In black and font type 'NimbusSanL-Regu' is the original referee comment
In blue and font type 'NimbusSanL-Regu' is our response to the referee comment
In black and font type 'Times New Roman' is unchanged text from the manuscript
In red and font type 'Times New Roman' is the text added to the manuscript to address the referee comment.

Referee 1 comment 1:
Validating satellite soil moisture product is necessary. However, this study focused on a very specific region (one SMAP pixel), which absolutely limits the value of this study. The authors need to clarify how such one-pixel evaluation can advance the understanding of satellite observed soil moisture.

Authors' response:
The authors agree with the referee that accuracy assessment of satellite soil moisture products is ideally performed using independent references collected from as many places around the world as possible. The SMAP Cal/Val team has presented the worldwide assessment of the passive-only SMAP soil moisture products in, for instance, Colliander et al. (2017) and Chan et al. (2018). Both are cited in the introduction.

In this manuscript, we report on the validation of the SMAP passive-only product for one of the sites that has also been used by the SMAP Cal/Val team in their worldwide assessments. The assessment presented here is done over a longer time period and covers a wider spectrum of hydrometeorological conditions than in the aforementioned studies, ranging from very wet to very dry and from frozen to hot. In the revision, we added the following to highlight the difference with previous assessments presented in Colliander et al. (2017) and Chan et al. (2018), on P3L14-17:

Validation of the SMAP L2 passive-only soil moisture product is an extension of the earlier assessments presented in, for instance, Colliander et al. (2017) and Chan et al. (2018) in the sense that comparisons for the Twente site are presented for multiple years covering a wider spectrum of hydro-meteorological conditions ranging from cold wet winters to the dry hot European summer of 2018.

Moreover, this is not only a validation study. It also deals with the difficulties involved in the creation of consistent references for the assessment of satellite observed soil moisture, such as

data gaps in the records of individual measurement locations and spatial mismatch errors. Part of this analysis included the use of soil moisture simulation by the Dutch integrated water resources model, which does not run on a global domain. This was explained in the submitted manuscript now around P3L814:

In this paper, we report on the development of a model-based upscaling method with scaling parameters derived directly from the mean and standard deviation of the *in situ* measured and simulated soil moisture. We adopted the Dutch integrated water resources model (De Lange et al. 2014), called 'Landelijk Hydrologisch Model' (LHM, National Hydrological Model in Dutch) that simulates the transfer of water masses across the groundwater, unsaturated zone and surface water reservoirs. LHM simulated soil moisture matching *in situ* measurements from January 2015 till October 2018 were utilized to develop the upscaling functions, which were subsequently used to assess the performance of the SMAP L2 passive-only soil moisture product for the period from April 2015 to December 2018.

To better highlight the upscaling dealt with in the manuscript, we have included the word 'upscaled' in the title and changed it to

Validation of SMAP L2 passive-only soil moisture products using upscaled *in situ* measurements collected in Twente, The Netherlands

Considering all this, we believe that the value of this research lies in the identification of factors that contribute to the differences found between an *in situ* reference and the satellite observed soil moisture. Indeed, the identification is done for a specific region, but the found contributing factors can be linked to hydrometeorological process, which are universal and occur all around the world. We provide evidence that large mismatches between *in situ* reference and SMAP soil moisture can be attributed to situations with strong vertical dielectric gradients found at the onset of soil freezing or wetting. This knowledge can help us to improve the current products and to make better use of the available products.

References:
Chan, S.K., Bindlish, R., O'Neill, P., Jackson, T., Njoku, E., Dunbar, R.S., Chaubell, J., Piepmeier, J., Yueh, S., Entekhabi, D., Colliander, A., Chen, F., Cosh, M.H., Caldwell, T.G., Walker, J., Berg, A.A., McNairn, H., Thibeault, M., Martínez-Fernández, J., Uldall, F., Seyfried, M., Bosch, D.D., Starks, P.J., Holifield-Collins, C.D., Prueger, J.H., van der Velde, R., Asanuma, J., Palecki, M., Small, E.E., Zreda, M., Calvet, J.C., Crow, W.T. and Kerr, Y.H.: Development and assessment of the SMAP enhanced passive soil moisture product, Remote Sens. Environ., 204, 931-941, doi: 10.16/j.rse.2017.08.025, 2018.

Colliander, A., Jackson, T.J., Bindlish, R., Chan, S., Das, N., Kim, S.B., Cosh, M.H., Dunbar, R.S,. Dang, L., Pashaian, L., Asanuma, J., Aida, K., Berg, A., Rowlandson, T., Bosch, D.D., Caldwell, T., Caylor, K., Goodrich, D.C., Al Jassar, H., Lopez-Baeza, E., Martinez-Fernandez, J., Gonzalez-Zamora, A., Livingston, S., McNairn, H., Pacheco-Vega, A., Moghaddam, M., Montzka, C., Notarnicola, C., Niedrist, G., Pellarin, T., Prueger, J., Pulliainen, J., Rautiainen, K., Garcia-Ramos, J.V., Seyfried, M., Starks, P.J., Su, Z., Zeng, Y., van der Velde, R., Thibeault, M., Dorigo, W.A., Vreugdenhil, J.M., Walker, J.P., Wu, X., Monerris, A., O'Neill, P.E., Entekhabi, D., Njoku, E.G., and

Yueh, S.: Validation of SMAP surface soil moisture products with core validation sites, Remote Sens. Environ., 191, 215-231, doi:10.1016/j.rse.2017.01.021, 2017.

Referee 1 comment 2:
If I understand correctly, the upscaling method used in this study is standardizing the model simulation by in situ observation. I would like to see how much improvement has been made by incorporating in situ values. If the improvement is tiny, then the contribution of in situ data is negligible. It doesn't make sense to assume model simulation as ground truth and to use it to validate other observations.

Authors' response:
Actually, we use the model output to create upscaling functions to translate the spatial mean of point measurements to the domain of the SMAP reference pixel. In all cases the upscaling functions are applied to the *in situ* measurements and the model simulations are never assumed to be the ground truth.

In the revision we will put an emphasis on clarifying that the SMAP retrievals are validated in all cases using *in situ* measurements. The following changes have been made.

In the abstract on P1L17-20:

The native and upscaled spatial soil moisture means computed using the *in situ* measurements have been adopted as references to assess the performance of the SMAP i) Single Channel Algorithm at Horizontal Polarization (SCA-H), ii) Single Channel Algorithm at Vertical Polarization (SCA-V), and iii) Dual Channel Algorithm (DCA) soil moisture estimates.

The introduction around P3L12-14

LHM simulated soil moisture matching *in situ* measurements from January 2015 till October 2018 were utilized to develop the upscaling functions. These functions were applied to upscale the *in situ* measurements used to assess the performance of the SMAP L2 passive-only soil moisture product for the period from April 2015 to December 2018.

The conclusion around P16L16-18:

We have adopted the native and upscaled spatial mean calculated using the *in situ* soil moisture measurements as references to assess the SMAP soil moisture estimates obtained with the i) Single Channel Algorithm at Horizontal polarization (SCA-H), ii) Single Channel Algorithm at Vertical polarization (SCA-V) and iii) Dual Channel Algorithm (DCA).

Referee 1 comment 3:
The authors keep using the model simulated root zone soil moisture. Please clarify why you don't use top 5-cm soil moisture from the model.

Authors' response:

We use the model simulated root zone soil moisture for developing the upscaling function and apply the developed functions to the soil moisture measured in situ at a 5 cm depth. The root zone soil moisture is used in this investigation because this is the shallowest soil layer for which the model (LHM) provides soil moisture contents.

Of course, the 5 cm and root zone soil moisture are not the same. In chapter 5, however, we demonstrate that a linear relationship exists between 5 cm in situ measured soil moisture and the model simulated root zone values. The model simulated root zone soil moisture is linearly transformed to match 5 cm in situ measurements using the obtained relationships.

The reason for selecting this model is because it is the Dutch national hydrological model that couples physically-based modelling approach for the unsaturated, groundwater and surface water flow. In particular, the first and the second are important in regions with shallow groundwater tables, such as the Netherlands.

It should, however, be emphasized that the model simulated root zone soil moisture is only used for the development of spatial upscaling functions and that the validation is still performed with the *in situ* soil moisture measured at a soil depth of 5 cm. We trust that the changes made in response to your previous comment (referee 1 comment 2) have clarified this.

Referee 1 comment 4:
Please describe the uncertainties from in situ measurements and discuss how these uncertainties will influence the findings.

Authors' response:
Section 2.2 describes the Twente measurement network and along with it the measurement uncertainty. This is estimated at 0.023 $m^3 m^{-3}$ and 0.027 $m^3 m^{-3}$ for the EC-TM and 5TM probes with the soil specific calibration function developed under laboratory conditions, see P4L19 and P4L28. *In situ* measurements from individual stations also include uncertainties due to spatial scale mismatch. In this research we considered this spatial-scale mismatch uncertainty by 1) taking the mean of a number of independent samples, and 2) developing upscaling function using spatially distributed model simulations.

The measurement uncertainty will affect the findings in such way that the larger the number of independent samples used for determining the spatial mean the smaller the effect of the *in situ* measurement uncertainty will be on the overall error metrics. This is discussed between p13l11 and p13l16 to which we have added,

'This is somewhat counterintuitive as soil moisture data from on average 11.0 and 13.7 stations contribute to the all-station means of pixels 3306/3606 and pixel 4371, respectively, and a large number of samples would implicate that the inherent measurement uncertainty (see Sect. 2.2) contribution to the total uncertainty of spatial mean reduces. Apparently, the increment from 7 to 11.0 or 13.7 samples is insufficient to lower the inherent measurement uncertainty is such way that it affects total uncertainty of the spatial mean significantly.'

Other specific comments:
Referee 1 specific comment 1: Title: "the Netherlands".

Authors' response: done

Referee 1 specific comment 2: Line 7: "RMSE".
Authors' response: done

Referee 1 specific comment 3: Table 2: what does "#" mean?
Authors' response: we will replace # with -

Referee 1 specific comment 4: Table 3: reformat the table.
Authors' response: we reformated the table so that width of columns is more appropriate for text, but please note that this is subject to typesetting.

Referee 1 specific comment 5: Figure 8: enlarge the temperature and precipitation.

Authors' response: we enlarged the temperature and precipitation plots in figure 8.

**Anonymous Referee 2**

The paper compares SMAP soil moisture estimates with in situ measurements for validation purposes, which is a rather common practice with satellite-derived products. Besides all the technical details and difficulties in the process due to representativeness and scale mismatch, I found that the most interesting (and potentially novel) piece of research consists of the use of a model to get more spatially distributed data and upscale the in situ measurements to the scale of the satellite pixel size. Unfortunately, the upscaling does not seem to produce better results than the raw in situ measurements when compared to the SMAP values. Still, the results are interesting.

Authors' response:
The authors would like to thank the referee for carefully reading our manuscript and providing constructive criticisms. In our response below we reply to the individual concerns.

In black and font type 'NimbusSanL-Regu' is the original referee comment.
In blue and font type 'NimbusSanL-Regu' is our response to the referee comment.
In black and font type 'Times New Roman' is unchanged text from the manuscript.
In red and font type 'Times New Roman' is the text added to the manuscript to address the referee comment.

Referee 2 comment 1:
I believe the authors should explain better why they used a model for root-zone soil moisture (40 cm depth) to represent the 5 cm depth soil moisture (both in situ and SMAP estimates).

Authors' response:
The model used in this study is the Dutch national hydrological model (LHM). This is a modelling framework that couples physically-based modelling approaches for groundwater, unsaturated soil and surface water flow. Particularly the combination of groundwater and unsaturated soil water flow is advantageous for a region with shallow groundwater tables, such as the Netherlands.

Another advantage of using LHM is that it makes use of best possible boundary conditions and atmospheric forcings, e.g. 100 m resolution subsurface information, 500 m resolution soil maps and coupled soil physical characteristics, 5 m resolution Digital Terrain Model, 25 m resolution land use map, 1 km resolution daily rainfall and reference evapotranspiration.

We agree with the reviewer that for this research a model with a shallower top soil layer would have been better, but that would have been impossible with the set of boundary conditions and atmospheric forcings available for the LHM. Therefore, we have chosen to use the root zone soil moisture simulated by the LHM and transform this towards the statistical moments of the 5 cm in situ measurements as is presented in section 5.1. We justify the validity of this approach based on the findings of Caranza et al. (2018). They found strong linear relationships between the 5 cm and 40 cm *in situ* measured soil moisture by the Twente monitoring network suggesting that the surface soil moisture can be used proxy for the root zone soil moisture and vice versa. Pezij et al. (2019) have previously adopted this assumption successfully for the

assimilation of the SMAP L3 product into the LHM. Here we used it to develop upscaling functions to translate the spatial mean of point measurements to the domain of the SMAP reference pixels.

Section 5.1 is fully devoted to the justification for using the simulated root zone soil moisture as proxy of the *in situ* measured soil moisture at 5 cm depth. A discussion on this topic in the context of Carranza et al. (2018) and Pezij et al. (2019) can be found around P10L16-21.

In revised manuscript we have referred to this discussion when we introduce the use of the LHM root zone soil moisture simulations around p6l25-31 via,

'The model's ability to provide the root zone soil moisture content as shallowest soil layer is clearly suboptimal for its use in the development of upscaling functions for the 5 cm *in situ* measurements. Nevertheless, we have chosen to use LHM simulations for the development of the upscaling functions for two main reasons. Firstly, LHM is an integrated hydrological model that couples unsaturated and groundwater flow processes, which is important for our study area where the groundwater tables are shallow. Secondly, the LHM makes use of the best possible land surface parameterization and atmospheric forcings, see above description, which is essential for a proper characterisation of the spatial heterogeneity. In section 5, we elaborate further on how the LHM root zone simulations are adopted for upscaling 5 cm soil moisture measurements.'

Further, it should also be noted that the soil moisture states in process models are defined at the mid-point of the soil layer, 20 cm, and the 5TM probe installed 5 cm below the soil surface has an approximate 4 cm influence zone and measures over a 1-9 cm soil layer. The text is updated with the fact that the model states are defined at the mid-point of the root zone layer, i.e. 20 cm, around p9l23-25.

'This may be attributed to the difference in soil layer thickness for which information is provided. In the case of the *in situ* measurements, the probes have an 4 cm influence zone (e.g. Benninga et al. 2018) and, thus, provide information for the 1 - 9 cm soil layer, while the LHM root zone layer has for the Twente region a nominal depth of 40 cm with the moisture state defined at the mid-point of the layer at 20 cm.'

References:
Carranza, C. D. U., van der Ploeg, M. J., and Torfs, P. J. J. F.: Using lagged dependence to identify (de)coupled surface and subsurface soil moisture values, Hydrol. Earth Syst. Sci., 22, 2255-2267, doi:10.5194/hess-22-2255-2018, 2018.

Pezij, M., Augustijn, D.C.M., Hendriks, D.M.D., Weerts, A.H., Hummel, S., van der Velde, R., and Hulscher, S.J.M.H.: State updating of root zone soil moisture estimates of an unsaturated zone metamodel for operational water resources management, J. Hydrol. X, 4, 100040, doi: 10.1016/j.hydroa.2019.100040, 2019

Referee 2 comment 2:
Figures 5 and 6 shows that the match between model and measurements is not very good and I was wondering if the authors tried to get a relationship between the 5 cm and the 40 cm soil moisture values.

Authors' response:

Indeed the 1:1 match between the measurements and the model simulations has imperfections, but we can also note that both respond in a similar fashion to rainfall inputs as is supported by the high correlation coefficients (> 0.88). In the manuscript in section 5.1 we attribute the difference between the 5 cm in situ measured soil moisture and the root zone soil moisture simulations to discrepancies in soil properties, in soil depth over which information is provided and the decoupling of the surface and subsurface soil moisture when the groundwater retreats.

For this manuscript, we have made use of the complete matchup set as well as the part of the set for which a linear relationship is found to develop the upscaling functions. We have not included a comparison between the root zone soil moisture simulations and the subsurface *in situ* measurements because we do not find this relevant for the objectives of this research. Notably, the purpose of using LHM root zone soil moisture simulations is primarily for the development of the upscaling functions for the top 5 cm soil moisture.

However, we do find the question interesting and for the response to this referee comment, we have made the comparison between the LHM root zone soil moisture and 20 cm *in situ* measurements. Note that we have taken the 20 cm measurements because this is at the center of the 40 cm root zone layer in LHM and the model states are defined at the mid-point of the soil layer. The matchups are shown in the figure 1 and the metrics obtained are: a correlation coefficient of 0.95, a RMSE of 0.066 $m^3$ $m^{-3}$ and a bias of 0.067 $m^3$ $m^{-3}$. In general, the distribution of the data points in the plot are similar to those in figure 6 except that the spread of the data points is somewhat smaller and discontinuity is less sharp than in the comparison with the *in situ* soil moisture measured at a 5 cm depth.

For the comparison of the surface and subsurface soil moisture the referee is referred to Carranza et al. (2018).

[Figure]

Figure 1. LHM root zone soil moisture against *in situ* measured soil moisture at a depth of 20 cm.

Referee 2 comment 3:
The validation is then carried out using the raw in situ measurements because it gives better results.

Authors' response:
This is not correct. We have carried out the validation with two references based on the native spatial mean derived from the *in situ* measurements and four references based on different versions of the upscaled spatial mean also derived from the *in situ* measurements. The results showed, however, that the best error metrics are obtained with the reference for which no upscaling has been performed.

In the discussion, section 7, we further analyse the error distribution and hydrometeorological circumstances under which large errors occur. We have chosen to only present the results with the native spatial mean for brevity. This has no influence on the findings in general because the upscaling is linear.

Referee 2 comment 4:
The authors explain mismatches based on physical processes and the inability of SMAP to capture those processes. I would have liked to see more references when describing potential sources of error.

Authors' response:
In the section 7.2 we refer to Colliander et al. (2017), Zheng et al. (2019) and Shellito et al. (2016) in the context of the dependence of the sampling depth on the soil moisture content. We

have added Escorihuela et al. (2010) as additional reference supporting the dependence of the sampling depth on the soil moisture content.

Further we have expanded the discussion on effect of standing water (floods) on microwave radiometry by adding the following around P14L31-P15L3,

'However, the winter 2015/16 overestimation was not preceded by a drought. In fact, it was quite wet (see Fig. 1) with small scale flooding on agricultural parcels across the Twente region. Since it is well known that standing water lowers the L-band emissivity, we expect that this contributed to SMAP's overestimation during the winter of 2015/16. Gouweleeuw et al. (2012) and Ye et al. (2015) have indeed investigated the soil moisture overestimation by microwave radiometry as a result of standing water. Researchers (e.g. Du et al. 2018, Schroeder et al. 2014) have even been using microwave radiometry for worldwide assessments of the faction of land covered by water.'

In the context of the effect of soil freezing on SMAP soil moisture estimates we revised the text around p15l4-l7 as follows,

As suggested in Sect. 6.1, the large underestimations by SMAP can often be associated with frozen conditions. Notably, when the water molecules are bound, as in ice, the dielectric constant of the medium reduces to levels comparable to that of dry soil (Rautiainen et al. 2014, Mironov et al. 2017) and, therefore, the SMAP estimates decrease as the ice content in the soil increases.

References:
Colliander, A., Jackson, T.J., Bindlish, R., Chan, S., Das, N., Kim, S.B., Cosh, M.H., Dunbar, R.S,. Dang, L., Pashaian, L., Asanuma, J., Aida, K., Berg, A., Rowlandson, T., Bosch, D.D., Caldwell, T., Caylor, K., Goodrich, D.C., Al Jassar, H., Lopez-Baeza, E., Martinez-Fernandez, J., Gonzalez-Zamora, A., Livingston, S., McNairn, H., Pacheco-Vega, A., Moghaddam, M., Montzka, C., Notarnicola, C., Niedrist, G., Pellarin, T., Prueger, J., Pulliainen, J., Rautiainen, K., Garcia-Ramos, J.V., Seyfried, M., Starks, P.J., Su, Z., Zeng, Y., van der Velde, R., Thibeault, M., Dorigo, W.A., Vreugdenhil, J.M., Walker, J.P., Wu, X., Monerris, A., O'Neill, P.E., Entekhabi, D., Njoku, E.G., and Yueh, S.: Validation of SMAP surface soil moisture products with core validation sites, Remote Sens. Environ., 191, 215-231, doi:10.1016/j.rse.2017.01.021, 2017.
Du, J., Kimball, J.S., Galantowicz, J., Kim, S.-B., Chan, S.K., Reichle, R., Jones, L.A., and Watts. J.D.: Assessing global surface water inundation dynamics from SMAP, AMSR2 and Landsat, Remote Sens. Environ., 213, 1-17, doi: 10.1016/j.rse.2018.04.054, 2018.
Gouweleeuw, B.T., van Dijk, A.I.J.M., Guerschman, J.P., Dyce, P., and Owe, M.: Space-based passive microwave soil moisture retrievals and the correction for a dynamic open water fraction, Hydrol. Earth Syst. Sci., 16, 1635-1645, doi:10.5194/hess-16-1635-2012, 2012.
Ye, N., Walker, J.P., Guerschman, J., Ryu, D., and Gurney, R.J.: Standing water effect on soil moisture retrieval from L-band passive microwave observations,  Remote Sens. Environ., 169, 232-242, doi: 10.1016/j.rse.2015.08.013, 2015.
Mironov, V.L., Kosolapoca, L.G., Lukin, Y.I., Karavaysky, A.Y., Molostov, I.P.: Temperature- and texture-dependent dielectric model for frozen and thawed minerals soils at a frequency of 1.4 GHz, Remote Sens. Environ., 200, 240-249, doi: 10.1016/j.rse.2017.08.007, 2017.
Rautiainen, K., Lemmetyinnen, J., Schwank, M., Kontu, A., Ménard, C.B., Mätzler, C., Drusch, M., Wiesmann, A., Ikonen, J., and Pulliainen, J.: Detection of soil freezing from L-band passive

microwave observations, Remote Sens. Environ., 147, 206-218, doi: 10.1016/j.rse.2014.03.007, 2014.

Schroeder, R., McDonald, K.C., Chapman, B.D., Jensen, K., Podest, E., Tessler, Z.D., Bohn, T.J., and Zimmermann, R.: Development and evaluation of a multi-year factional surface water data set derived from active and passive microwave remote sensing data, Remote Sens., 7, 16688-16731, doi:10.3390.rs71215843, 2015.

Shellito, P.J., Small, E.E., Colliander, A., Bindlish, R., Cosh, M.H., Berg, A.A., Bosch, D.D., Caldwell, T.G., Goodrich, D.C., McNairn, H., Prueger, J.H., Starks, P.J., van der Velde, R. and Walker, J.P.: SMAP soil moisture drying more rapid than observed in situ following rainfall events, Geophys. Res. Lett., 43. 9068-8075, doi: 10.1002/2016/GL069946, 2016.

Zheng, D., Li, X., Wang, X., Wang, Z., Wen, J., van der Velde, R., Schwank, M., and Su, Z.: Sampling depth of L-band radiometer measurements of soil moisture and freeze-thaw dynamics on the Tibetan Plateau. Remote Sens. Environ., 226, 16-25, doi: 10.1016/j.rse.2019.03.029, 2019.

Referee 2 comment 5:
The last paragraph does not seem to follow directly from the objectives of the paper or the results presented in the paper. It is also not clear to me how the mismatches can be used to help management.

Authors' response:
We agree with the referee that the last paragraph does not follow directly from the results of the paper and has been removed from the manuscript.

Referee 2 comment 6:
My last point is regarding the abstract: even though the fact that the upscaling did not work is included in the conclusions, it has not been included in the abstract.

Authors' response:
Our intention was to keep the abstract short with only the most relevant information for the readers. In the abstract we do mention around l16-17 the use of the Dutch national hydrological model for 'the development of upscaling functions to translate the spatial mean of point measurements to the domain of the SMAP reference pixels'. But indeed we do not follow up on the results obtained as we do in the conclusion with 'The upscaled in situ reference do not result in better metrics'. In response to this comment we have modified as abstracts as follows around P1L21-24,

'In the case of the Twente network it was found that the SCA-V soil moisture retrieved SMAP observations collected in the afternoon had the best agreement with the native spatial mean leading to an unbiased Root Mean Squared Error (uRMSE) of 0.059 $m^3$ $m^{-3}$, whereas for the upscaled *in situ* references primarily larger biases were found. These error levels are larger than the mission's target accuracy of 0.04 $m^3$ $m^{-3}$, which can be attributed to large over- and underestimation errors ($>0.08$ $m^3$ $m^{-3}$) in particular at the end of dry spells and during freezing, respectively.'

Referee 2 comment 7:
A minor point is that in the supplement, the tittle of the paper wrongly includes the word "upscale".

Authors' response: Thank you, we decide based on the comments of referee 1# to include the word 'upscaled' in the title.